# *Bacteroides* vesicles promote functional alterations in the gut microbiota composition

Olga Yu. Shagaleeva,[1] Daria A. Kashatnikova,[1,2] Dmitry A. Kardonsky,[1] Boris A. Efimov,[1,3] Viktor A. Ivanov,[1] Svetlana V. Smirnova,[2] Suleiman S. Evsiev,[1] Eugene A. Zubkov,[4] Olga V. Abramova,[4] Yana A. Zorkina,[4] Anna Y. Morozova,[4] Elizaveta A. Vorobeva,[1] Artemiy S. Silantiev,[1] Irina V. Kolesnikova,[1] Maria I. Markelova,[5] Evgenii I. Olekhnovich,[1] Maxim D. Morozov,[1] Polina Y. Zoruk,[1] Daria I. Boldyreva,[1] Victoriia D. Kazakova,[1] Anna A. Vanyushkina,[6] Andrei V. Chaplin,[1,3] Tatiana V. Grigoryeva,[5] Natalya B. Zakharzhevskaya[1]

**ABSTRACT** Inflammatory bowel diseases are characterized by chronic intestinal inflammation and alterations in the gut microbiota composition. *Bacteroides fragilis*, which secretes outer membrane vesicles (OMVs) with polysaccharide A (PSA), can moderate the inflammatory response and possibly alter the microbiota composition. In this study, we created a murine model of chronic sodium dextran sulfate (DSS)-induced intestinal colitis and treated it with *B. fragilis* OMVs. We monitored the efficiency of OMV therapy by determining the disease activity index (DAI) and performing histological examination (HE) of the intestine before and after vesicle exposure. We also analyzed the microbiota composition using 16S rRNA gene sequencing. Finally, we evaluated the volatile compound composition in the animals' stools by HS-GC/MS to assess the functional activity of the microbiota. We observed more effective intestinal repair after OMV treatment according to the DAI and HE. A metabolomic study also revealed changes in the functional activity of the microbiota, with a predominance of phenol and pentanoic acid in the control group compared to the group treated with DSS and the group treated with OMVs (DSS OMVs). We also observed a positive correlation of these metabolites with *Saccharibacteria* and *Acetivibrio* in the control group, whereas in the DSS group, there was a negative correlation of phenol and pentanoic acid with *Lactococcus* and *Romboutsia*. According to the metabolome and sequencing data, the microbiota composition of the DSS-treated OMV group was intermediate between that of the control and DSS groups. OMVs not only have an anti-inflammatory effect but also contribute to the recovery of the microbiota composition.

**IMPORTANCE** *Bacteroides fragilis* vesicles contain superficially localized polysaccharide A (PSA), which has unique immune-modulating properties. Isolated PSA can prevent chemically induced colitis in a murine model. Outer membrane vesicles (OMVs) also contain digestive enzymes and volatile metabolites that can complement the anti-inflammatory properties of PSA. OMVs showed high therapeutic activity against sodium dextran sulfate-induced colitis, as confirmed by histological assays. 16S rRNA sequencing of fecal samples from different inflammatory stages, supplemented with comprehensive metabolome analysis of volatile compounds conducted by HS-GC/MS, revealed structural and functional alterations in the microbiota composition under the influence of OMVs. Correlation analysis of the OMV-treated and untreated experimental animal groups revealed associations of phenol and pentanoic acid with *Lactococcus*, *Romboutsia*, *Saccharibacteria*, and *Acetivibrio*.

**KEYWORDS** HS-GC/MS, IBD, OMVs, DSS-induced colitis

Address correspondence to Natalya B. Zakharzhevskaya, natazaha@gmail.com.

The authors declare no conflict of interest.

See the funding table on p. 15.

nflammatory bowel disease (IBD) is a heterogeneous chronic and relapsing inflammatory disease that is traditionally divided into ulcerative colitis (UC) and Crohn's disease (CD) (1–3). Both UC and CD can cause various symptoms, such as diarrhea, rectal bleeding, and abdominal pain. However, inflammation in CD is transmural and can affect any part of the gastrointestinal tract. In contrast, inflammation in UC is more superficial and limited to the colon (4). Although the pathogenesis of IBD remains unknown, it has been described as a multifactorial disease involving both genetic and environmental components (5). Changes in the gut microbiota have also been associated with IBD. Standard IBD therapy aims to reduce the acute phase of inflammation with hormones and antibiotics. At the same time, antibiotics cause changes in the microbiota composition, which contributes to increased inflammation (3, 6). This creates a vicious cycle and provokes the development of chronic inflammation in the gut. Current probiotics are largely ineffective because bacterial colonization occurs during the acute phase of inflammation, when survival conditions are extreme. Bacteria may exhibit new virulent properties when they adapt to chronic inflammation in the intestine (7). In this regard, it is important to select microorganisms that are able to suppress inflammation and promote effective colonization.

Among bacteria with immunomodulatory properties, *Bacteroides fragilis* is the most well-known because of the presence of surface polysaccharide A (PSA) (8). PSA interacts with TLR2 receptors on dendritic cells, leading to the synthesis of interleukin-10. Interleukin-10 (IL-10) modulates the activity of $CD4^+FOXP3+$ regulatory T cells (Tregs), which suppress inflammation (9, 10). PSA is also found on the surface of outer membrane vesicles (OMVs), which are produced in large quantities by *B. fragilis* (11). *B. fragilis* may be used as a mono probiotic therapy or as part of a complex probiotic in the future, but it is important to continue the research of its biological properties (12). However, the use of *B. fragilis* OMVs with PSA may be justified. Previously, it was shown that PSA prevents experimental colitis in a murine model (13). However, the efficacy of OMVs as therapeutic agents has not been fully evaluated. In our previous research, we performed a detailed proteomic and metabolomic study of OMVs produced by the nontoxigenic *B. fragilis* strain JIM10 (14). Combined with the HS-GC/MS data, the proteomic and metabolomic results of *B. fragilis* OMV analysis revealed that vesicles contain several enzymes, metabolites, and volatile compounds, especially fatty acids (15). Multicomponent OMVs are expected to be more effective than isolated PSAs in reducing inflammation and subsequent microbial colonization (16).

Models of DSS-induced colitis are the most widely used models in IBD research and are also optimal for studying therapeutic agents (17, 18). DSS-induced colitis can be characterized by histological analysis and measurement of the disease activity index (DAI) (19, 20). The DAI includes general characteristics such as stool consistency, including the presence of blood, animal behavior and weight (21, 22). Traditionally, it has been assumed that DSS does not alter the microbiota composition. However, a number of recent studies have suggested that DSS may indirectly affect species diversity. A recent study of the microbiota composition in a DSS-induced colitis model revealed that two species, *Duncaniella muricolitica* and *Alistipes okayasuensis*, were associated with worse disease outcomes after DSS treatment (23). These pathogenic taxa reproduced a severe DSS response in noncolonized germ-free mice, confirming the causal relationship between these species and the severity of DSS-induced colitis. Therefore, it is important to assess the microbiota composition when DSS is used. Metagenomic research is traditionally used to characterize microbiota diversity (24). Volatile metabolites released by bacteria can also be useful for assessing the functional activity of the microbiota (25).

In this study, we created a murine model of DSS-induced colitis and tested the use of OMVs from the *B. fragilis* JIM10 strain as a therapeutic agent for intestinal recovery. We evaluated the effectiveness of OMV therapy by determining the DAI and histological examination (HE) based on histological examination for all groups. In addition, we determined the relative amounts of volatile compounds by HS-GC/MS and analyzed the composition of the gut microbiota by 16S rRNA sequencing.

## MATERIALS AND METHODS

### Bacterial strain and growth conditions

*B. fragilis* (JIM10 strain—NZ_MBRB00000000.1) was lyophilized at −80°C in 20% (wt/vol) sucrose and 1% (wt/vol) gelatin solution. The strain was grown anaerobically on anaerobe basal agar (Oxoid) supplemented with 5% (vol/vol) defibrinated sheep blood under anaerobic conditions established by placing Anaerogen bags in anaerobic 3.5 L jars (Oxoid; Thermo Fisher Scientific, Inc.) or in anaerobic jars (Schuett-Biotec, Germany) at 37°C until the stationary phase. For liquid culture, a preculture of the JIM10 strain was grown anaerobically in Columbia broth base (Hi Media, India) at 37°C until the stationary phase. Curves of growth are controlled by measuring optical at 600 nm.

### OMV isolation

About 200 mL of a 24-h liquid bacterial culture of *B. fragilis* JIM10 was centrifuged at $4,500 \times g$ at 4°C. To remove residual cells, the supernatant was filtered through a 0.45-µm porous membrane. The filtrate was subjected to ultracentrifugation at $100,000 \times g$ for 2 h (Optima L-90K ultracentrifuge; Beckman Coulter). The supernatant was discarded, and the pellet was washed with sterile phosphate-buffered saline (PBS) and filtered through a sterile 0.2-µm pore polyvinylidene difluoride (PVDF) membrane (Miltenyi GV; Millipore). The ultracentrifugation procedure was repeated two times. The vesicle pellet was resuspended in distilled water or 150 mM NaCl (pH 6.5). The concentration of OMVs was quantified as the dry precipitate.

### Electron microscopy

OMVs were diluted to concentrations ranging from 2 to $5 \times 10^{11}$ particles/mL and prepared for TEM analysis as previously described (14). Images were obtained using a JEM-1400 (Jeol, Tokyo, Japan) transmission electron microscope equipped with a Rio-9 camera (Gatan Inc., Pleasanton, CA, USA) at 120 kV.

### Animals

Two-month-old (20–25 g) female C57BL/6 mice obtained from the Laboratory Animal Nursery (Pushchino, RAS, Moscow Region) were used in the experiment. The animals were randomly divided into three groups: the control group ($n = 10$), the sodium dextran sulfate (DSS) group ($n = 10$), and the sodium dextran sulfate group treated with outer membrane vesicles (DSS + OMV) group ($n = 10$). Each mouse in the DSS group was gavaged orally with 4% DSS for 5 days. Intestinal inflammation was maintained for 20 days in the DSS group. During this time, animals were alternately given water with or without DSS. Detailed description of the DSS exposure: Days 1–5, DSS and DSS + OMV groups receive 3% DSS; Days 6–10, water for all groups; Days 11–16, the DSS group receives DSS solution, the DSS + OMV group receives DSS solution and vesicles; and Days 17–20, the DSS group receives water, DSS + OMV receives water and vesicles. The mice were anesthetized with isoflurane (4%, 2 L/min) in a chamber until their toe reflex disappeared. The mice were then quickly sacrificed by decapitation. All animals in the DSS + OMV group were maintained under the same conditions as those in the DSS group until the 10th day. Starting on Day 10, animals with persistent inflammation were treated with OMVs (1 µg/kg). All mice in the DSS + OMV group were anesthetized with isoflurane (4%, 2 L/min) in a chamber until their toe reflex disappeared on Day 20. The mice were then quickly sacrificed by decapitation. The control group of mice received the same volume of normal saline for 20 days prior to euthanasia. Body weight, stool consistency, and the presence of occult blood were measured every 3 days. Stool samples were collected for subsequent metabolomic and 16S rRNA gene sequencing.

## Reagents

Dextran sulfate sodium salt, Mr ~40,000, Alfa Assar. N,O-Bis(trimethylsilyl)trifluoroaceta-mide (with 1% trimethylchlorosilane; vol/vol; lot no. B-023) and heptadecanoic acid (as an internal standard, IS, purity ≥98%; lot no. H3500) were purchased from Sigma−Aldrich (Saint Louis, USA). O-Methyl hydroxylamine hydrochloride (purity: 98.0%; lot no. 542171) was purchased from J&K Scientific Ltd. (Beijing, China). Pyridine (lot no. C10486013) was obtained from Macklin Biochemical Co., Ltd. (Shanghai, China). Chromatographic-grade methanol was purchased from Thermo Fisher Scientific (Waltham, USA). Pure water was obtained from Wahaha Company (Hangzhou, China).

## Clinical disease score

The DAI was estimated by the score of body weight loss (no weight loss: 0; 5–10% weight loss: 1; 11–15% weight loss: 2; 16–20% weight loss: 3; and 20% wt loss: 4), stool consistency (formed: 0; watery stool: 2) and the degree of stool occult blood (normal stool color: 0; reddish stool color: 2). DAI comparisons were conducted using the Kruskal-Wallis test for each day.

## Histology

Colon samples were cut lengthwise, washed with PBS, and rolled into a "Swiss roll" 20. The obtained material was fixed in 10% buffered formalin, pH 7.0–7.8, for 48 h and then placed in labeled histological cassettes with a liner to prevent unrolling. Histological processing was performed with isopropyl alcohol (Chimmed, Russia) and two changes of o-xylene (Chimmed, Russia). The processed samples were embedded in paraffin (BioVitrum, Russia). Sections 3.5 µm in thickness were cut on a Thermo HM 340E rotary microtome (Thermo Scientific, China, manufacturer), as this cutting plane was perpendicular to the axis of the folded sample. After drying, the sections were stained with hematoxylin-eosin (BioVitrum, Russia). Additional staining with Alcian blue, pH 3.0, was performed by contrasting the nuclei with Mayer's hematoxylin, which allowed visualization of mucosal goblet cells. The samples were evaluated by light microscopy at magnifications of ×4, ×10, and ×40 (Zeiss Primo Star, China). A ball-based scoring system was used to assess the severity of damage. A nonparametric Mann–Whitney test was conducted for the histological index.

## HS-GC/MS

For HS-GC/MS, fecal samples (all available sample volumes) were collected every 3 days from the control, DSS, and DSS + OMV groups. Stool samples (50–100 mg) plus 500 µL water samples were placed in 10 mL screw-capped vials for a Shimadzu HS-20 headspace extractor. First, 0.2 g of a salt mixture (ammonium sulfate and potassium dihydrogen phosphate at a ratio of 4:1) was added to increase the ionic strength of the solution. The following headspace extractor settings were used: oven temperature 80℃, sample line temperature 220℃, transfer line temperature 220℃, equilibration time 15 min, pressurization time 2 min, loading time 0.5 min, injection time 1 min, and needle rinsing time 7 min. Vials were sealed and analyzed on a Shimadzu QP2010 Ultra GC/MS with a Shimadzu HS-20 headspace extractor, a VF-WAXMS column 30 m in length, 0.25 mm in diameter, and 0.25 µm phase in thickness. The initial column temperature was 80℃, the heating rate was 20 ℃/min to 240℃, and the exposure time was 20 min. The carrier gas used was helium (99.9999), the injection mode was splitless, and the flow rate was 1 mL/min. The ion source temperature was 230℃. And the interface temperature was 240℃. Total ion current (TIC) monitoring mode was used. The NIST 2014 Mass Spectra Library with Automated Mass Spectra Deconvolution and Identification System (AMDIS version 2.72) was used to analyze the obtained mass spectra.

## Metabolome data processing

The HS-GC/MS data were processed as follows: the peak areas calculated by AMDIS for the selected compounds were converted into relative abundances. Volatile compound percentages were estimated by summing the percentages of confidently identified compounds for each sample in the AMDIS database. The resulting values were then recalculated as a percentage of the total number of compounds confidently identified. These conversions were necessary to avoid errors in content estimation due to unreliable matrix signals caused by noise. The GC/MS data were processed using MetaboAnalyst 5.0 software (http://www.metaboanalyst.ca) and GraphPad Prism 8.0.1 software. The values obtained for each animal were considered paired and consistent, as confirmed by the ROUT outlier test [ROUT ($Q = 1\%$)]. As it was not possible to assess the normality of the distribution for all sample groups, it was assumed that the raw data did not conform to a normal distribution. The nonparametric Mann−Whitney test was used for the primary comparisons between groups. After natural log transformation, the normalized data were analyzed using a standard $t$ test and ANOVA. Statistical significance was determined by a two-tailed $P$ value of less than 0.05. Unsupervised principal component analysis (PCA) was used to reduce the number of dimensions and further explore the data. Standard procedures were applied prior to PCA. A chi-squared test was used to determine a method for imputing missing values. The missing at random (MAR) criteria were identified in the data. The BPCA method was found to be the most appropriate. Normalization and scaling procedures were then performed. Correlations between metabolites and gut microbiota composition were analyzed using the R "psych" package based on Spearman's rank correlation coefficient (26).

## DNA extraction

Fecal samples were used for total DNA extraction. Nucleic acids were extracted using the MagicPure Stool and Soil Genomic DNA Kit and Kingfisher Flex Purification System (Thermo Fisher Scientific, USA) according to the manufacturer's protocol. DNA was then quantified using the Quant-iT dsDNA BR Assay Kit (Thermo Fisher Scientific, USA) on a Qubit 4 fluorometer.

## 16S rRNA gene sequencing on the MinION platform

The extracted DNA (1–5 ng) was amplified using 27F (AGAGTTTGATYMTGGCTCAG) and 1492R (GGTTACCTTGTTAYGACTT) primers (Eurogen, Russia) and the Tersus Plus PCR Kit (Eurogen, Russia) in a total volume of 25 µL. Amplification was performed with the following PCR conditions: initial denaturation at 95°C for 2 min, 95°C for 1 min, 60°C for 1 min, and 72°C for 3 min; 27 cycles; and a final extension at 72°C for 2 min and 4°C cooling. The quality of the amplicons was checked by electrophoresis in a 1.5% agarose gel. The final amplicons were purified using KAPA HyperPure Beads (Roche, Switzerland) according to the manufacturer's protocol.

Libraries were prepared according to the manufacturer's protocol (ligation sequencing amplicons) with modifications. Amplicons were processed using the NEBNext Ultra II End Repair/dA-Tailing Module (NEB). Barcodes [Native Barcoding Kit 96 (SQK-NBD109.96)] were ligated using Blunt/TA Ligase Master Mil (NEB). Barcoded libraries were purified using KAPA Pure Beads (Roche, Switzerland). Library concentrations were measured using the Quant-iT dsDNA Assay Kit, High Sensitivity (Thermo Fisher Scientific, USA), and samples were mixed at equimolar concentrations. The final adapter [Adapter Mix II Expansion (Oxford Nanopore Technologies, UK)] was ligated to the pooled library using the NEBNext Quick Ligation Module (NEB). The prepared DNA library (12 µL) was mixed with 37.5 µL sequencing buffer and 25.5 µL loading beads, loaded onto an R9.4 flow cell (FLO-MIN106; Oxford Nanopore Technologies) and sequenced using MinION Mk1B. MINKNOW software ver. 22.12.7 (Oxford Nanopore Technologies) was used for data acquisition.

## Genome data processing

Technical sequences and bases with a quality lower than a Phred score of 9 were processed using Porechop V0.2.3 NanoFilt V2.8.0 software (27, 28). The resulting data were evaluated using the Emu pipeline for taxonomic classification (29). Alpha and beta diversity analyses were performed using the vegan package for GNU/R (30). Alpha and beta diversity were assessed using the Bioconductor Microbiota Process package for GNU/R. Heatmap visualization was performed using the "pheatmap" package for GNU/R. Alpha diversity studies were performed using the Microbiota Process package (31, 32).

## RESULTS

### Histological evaluation of intestinal inflammation and OMV therapy

The isolated OMVs were visually characterized by TEM. As previously described, the vesicles were approximately spherical, and most of the particles in the center were relatively dark (Fig. S1) (14). The obtained OMVs were used to treat mice after DSS exposure. All three groups were subjected to histological examination of intestinal tissues on the 10th and 20th days of the experiment. According to the histopathological results (Fig. 1A), the mucosa and submucosa of the colon in the control group (Fig. 1A K1-K2) were thin without signs of edema. The mucosa did not have extensive areas of damage along its entire length and retained smooth, slit-like crypts. The blood and lymphatic vessels were not dilated. The epithelium with mild inflammatory changes in several animals from the control group was characterized by nonextended areas of goblet cell loss (Fig. 1A, K3) (Fig. S2). In the DSS group (Fig. 1A, DSS 1–2), the structure of the mucosa in the distal parts of the colon was not determined; only individual goblet cells were observed (Fig. 1A, DSS 3) (Fig. S3). In other parts of the colon, focal lesions of the mucosa were observed to a lesser extent. The lesions showed loss of normal epithelial structure, dilated, branched crypts of reduced depth and abundant immune cell infiltration. Crypt abscesses and foci of mucosal infiltration were identified in the proximal intestine. An accumulation of immune cells was observed in the submucosal layer of the colon of all the experimental animals. The mucosa and submucosa were slightly thickened, and the blood and lymphatic vessels were dilated. A smaller lesion area was observed in the DSS-treated OMV group than in the DSS group (Fig. 1A, DSS-treated OMVs 1–2) (Fig. S4). Mucosal foci were also identified in the distal sections of the colon, but their structure was indistinguishable, and visually the number of goblet cells was reduced (Fig. 1A, DSS OMV 3). An accurate count of goblet cells was not carried out, and the assessment was subjective and exclusively visual, since it was not possible to calculate the number of cells in the area of total lesion. However, in contrast to those in the DSS group, the lesions in the DSS OMV group were focal in nature and had a significantly shorter duration. In the distal parts of the bowel, the lesions occurred over a longer period of time and were also more pronounced. Crypt abscesses and areas of ulceration were also observed in both the distal and proximal bowel but only in a few animals in the DSS group. In areas of mucosal lesions, crypts were dilated, and the number of goblet cells was also visually reduced. Many nodules of lymphoid tissue were observed in the submucosal layer. The mucous and submucous membranes were slightly thickened, and the blood and lymphatic vessels were dilated. Thus, histological examination of the bowel samples from the control group showed no signs of a massive inflammatory process. The characteristic histological signs of colitis were not observed. In the DSS group, chronic colitis, including changes in the epithelium (increased thickness of the submucosal and mucosal layers of the bowel, expansion or complete disappearance of crypts, inflammatory infiltration of the mucosa and submucosal layer, decreased number of goblet cells, and increased number of lymph nodes in the submucosal layer), was observed. In the DSS-treated OMV group, characteristic histological signs of chronic colitis were also observed, including focal changes in the epithelium, but the extent and degree of manifestation were less than those in the DSS group, suggesting a reduction in disease severity. A histopathological

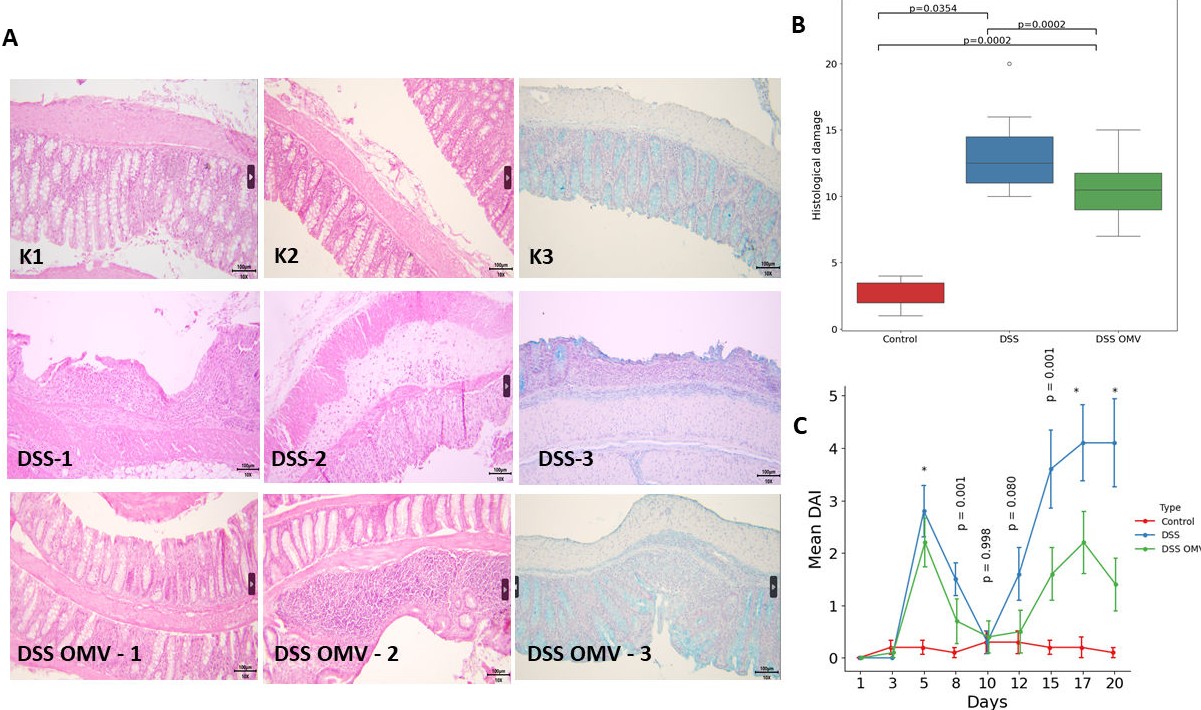

**FIG 1** Histological examination of intestinal tissue from all groups. (A) Formalin-fixed preparation of the distal intestine of an experimental animal. Hematoxylin-eosin staining (K1, K2, DSS1, DSS2, DSSOMV-1, and DSSOMV-2). Additional staining with Alcian blue, pH 3.0, was performed by contrasting the nuclei with Mayer's hematoxylin, which allowed visualization of mucosal goblet cells (K3, DSS3, and DSSOMV-3). K1, K2, K3—control (without DSS); DSS-1, DSS-2, DSS-3—DSS-treated intestine; DSSOMV-1, DSSOMV-2, DSSOMV-3—OMV-treated intestine. The samples were evaluated by light microscopy at magnifications of ×4, ×10, and ×40 (Zeiss Primo Star, China). (B) Evaluation of histological sections of the distal intestine at various time points. A nonparametric Mann–Whitney test was conducted for the histological index. (C) DAI for all three groups throughout the experiment, day by day. DAI comparisons were conducted using the Kruskal–Wallis test for each day. Values with "*" denote $P$-values < 0.001.

scoring system was used to assess epithelial damage and inflammatory cell infiltration. According to the data obtained (Table S1), the histological index was significantly lower in the DSS-treated OMV group than in the DSS group, indicating tissue repair under OMV treatment (Fig. 1B). As shown in Fig. 1C, the DAIs of the DSS and DSS OMV groups were significantly greater than those of the control group.

DAIs in both groups increased until Day 3, mainly due to weight loss, and peaked on Day 5, when severe diarrhea with visible blood in the stool was observed. DAIs decreased in both groups by Day 10 after the discontinuation of DSS. When DSS was resumed, an increased DAI was again observed in both groups. However, OMV therapy significantly decreased the DAI in the DSS-treated OMV group compared to the DSS group (Table S1).

## Metabolomic data

The metabolite spectrum was evaluated for samples collected on the 1st, 10th, and 20th days of the experiment. The study was performed by HS-GC/MS. The metabolomic data were normalized (Table S2). At least 60% of the detected stable compounds were used for the metabolomic profile comparison. The most frequently detected compounds meeting the screening criteria are listed in Table 1.

Among the compounds identified, short-chain, medium-chain, and long-chain fatty acids and amino acid derivatives were detected. Differences in the metabolic profiles of the control group and the DSS and DSS OMV groups were shown using PCA and OPLS-DA (Fig. 2A and B). According to the data obtained, the samples were clustered into three independent groups. The control group was distinguished by a separate characteristic spectrum of the relative abundance of volatile compounds in the DSS and

**TABLE 1** Complete list of stably detected metabolites on the three checkpoint days of the experiment

| Metabolite | Percent of samples (%) | | | | | | | | | Percent of samples in which the metabolite is detected $n = 263$ |
|---|---|---|---|---|---|---|---|---|---|---|
| | Control group (d) | | | DSS group (d) | | | DSS + OMV group (d) | | | |
| | 1 | 10 | 20 | 1 | 10 | 20 | 1 | 10 | 20 | |
| Acetic acid | 90.9 | 90.9 | 90.9 | 90.9 | 90.9 | 90.9 | 90.9 | 90.0 | 90.9 | 100.0 |
| Propanoic acid | 90.9 | 90.9 | 90.9 | 90.9 | 90.9 | 90.9 | 90.9 | 90.0 | 90.9 | 100.0 |
| Benzaldehyde | 90.9 | 72.7 | 90.9 | 72.7 | 72.7 | 72.7 | 72.7 | 50.0 | 63.6 | 79.8 |
| Propanoic acid, 2-methyl- | 90.9 | 90.9 | 90.9 | 90.9 | 90.9 | 90.9 | 90.9 | 90.0 | 90.9 | 100.0 |
| Benzeneacetaldehyde | 90.9 | 72.7 | 81.8 | 90.9 | 63.6 | 90.9 | 90.9 | 50.0 | 72.7 | 87.8 |
| Butanoic acid, 3-methyl- | 90.9 | 90.9 | 90.9 | 90.9 | 90.9 | 90.9 | 90.9 | 80.0 | 90.9 | 99.6 |
| Methyl formate | 63.6 | 81.8 | 72.7 | 81.8 | 63.6 | 54.5 | 90.9 | 50.0 | 81.8 | 72.2 |
| Butanoic acid, 2-methyl- | 90.9 | 90.9 | 90.9 | 90.9 | 81.8 | 81.8 | 90.9 | 90.0 | 90.9 | 96.6 |
| Pentanoic acid | 90.9 | 90.9 | 90.9 | 90.9 | 90.9 | 90.9 | 90.9 | 90.0 | 90.9 | 100.0 |
| Pentanoic acid, 4-methyl- | 72.7 | 90.9 | 81.8 | 90.9 | 54.5 | 72.7 | 90.9 | 20.0 | 45.5 | 77.6 |
| Phenol | 90.9 | 90.9 | 90.9 | 90.9 | 72.7 | 72.7 | 90.9 | 40.0 | 90.9 | 81.7 |
| Octanoic acid | 9.1 | 90.9 | 81.8 | 90.9 | 63.6 | 63.6 | 90.9 | 50.0 | 81.8 | 79.1 |
| Benzyl alcohol | 90.9 | 90.9 | 90.9 | 90.9 | 81.8 | 81.8 | 90.9 | 90.0 | 90.9 | 97.3 |
| Undecanal | 72.7 | 72.7 | 54.5 | 63.6 | 63.6 | 27.3 | 81.8 | 50.0 | 72.7 | 76.4 |
| Nonanoic acid | 63.6 | 90.9 | 90.9 | 90.9 | 45.5 | 90.9 | 90.9 | 80.0 | 81.8 | 89.4 |
| 5-Acetyl-2-methylpyridin | 81.8 | 72.7 | 54.5 | 54.5 | 72.7 | 27.3 | 63.6 | 70.0 | 54.5 | 67.7 |
| Indole | 90.9 | 90.9 | 81.8 | 90.9 | 81.8 | 72.7 | 72.7 | 80.0 | 72.7 | 90.9 |
| Tetradecanal | 90.9 | 72.7 | 54.5 | 81.8 | 9.1 | 63.6 | 72.7 | 40.0 | 45.5 | 64.6 |
| Benzenepropanoic acid | 36.4 | 63.6 | 81.8 | 72.7 | 9.1 | 81.8 | 90.9 | 30.0 | 90.9 | 63.5 |
| Heptanoic acid | 18.2 | 72.7 | 90.9 | 90.9 | 45.5 | 63.6 | 90.9 | 30.0 | 81.8 | 71.9 |
| Phenylethyl alcohol | 81.8 | 63.6 | 81.8 | 81.8 | 45.5 | 45.5 | 81.8 | 70.0 | 90.9 | 70.7 |

DSS OMV groups on the 10th day of the experiment. No significant differences were found between the DSS and DSS OMV groups on Day 10 of the experiment. On Day 20, corresponding to the end of the vesicle treatment, three separate groups could be distinguished. The control group was significantly different from the DSS and DSS OMV groups. A small number of metabolomic differences could be identified between the DSS and DSS OMV groups according to the PCA data. OPLS DA allowed the assessment of the magnitude of the differences, which contributed to a clearer classification of the DSS and DSS OMV groups.

The general profile of the DSS OMV group was characterized by 2,5-dimethyl-4-hydroxy-3(2H)-furanone, H-1-benzopyran-2-one, 3,4-dihydro, and nonane, which were not detected in the control or DSS groups (Fig. 2C). The overall profile of metabolites was significantly lower in the DSS group than in the control and DSS-treated OMV groups. When treated with vesicles, a tendency to restore the overall metabolite profile could be observed (Fig. 2C).

Significant changes in the relative amounts of metabolites were observed both on Day 20 when comparing the control and DSS OMV groups and when comparing the DSS and DSS OMV groups. The main changes included changes in the levels of acetic acid, propanoic acid, propanoic acid 2-methyl, benzaldehyde, pentanoic acid, phenol, nonanoic acid, heptanoic acid, and hexanoic acid (Fig. 3A and B). These metabolomic differences were identified in all three groups on the 20th day.

Most of the significant differences were found between the control and DSS-treated OMV groups on Day 20.

## Amplicon sequencing data analysis

Sequencing analysis of microbiome variability in stool samples was performed on the 1st, 10th, and 20th days of the experiment (Table S3). As shown in Fig. 4, at the

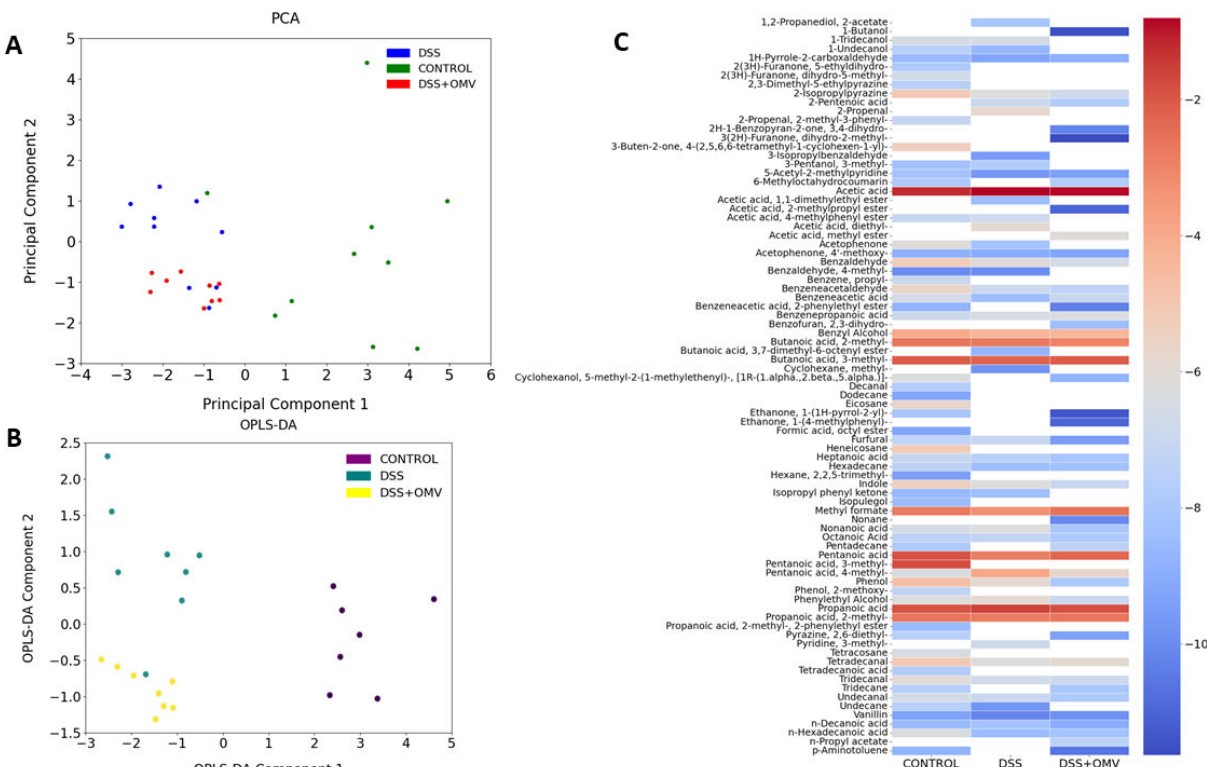

**FIG 2** Total HS-GC/MS data for all groups on the 20th day of the experiment. (A, B) PCA and OPLS-DA data represent three independent groups of samples according to the relative concentration of volatile compounds. (C) VOC (volatile compound) compounds identified in all groups. The relative concentrations in the vapor phase were used.

beginning of the experiment, homogeneous bacterial species were observed in the analyzed groups. At the same time, there was a significant predominance of Firmicutes in the microbiota. On the 10th day of the experiment, a decrease in the number of several bacterial species was observed in the DSS and DSS OMV groups compared to the control group. As expected, uniformity was still observed in the DSS and DSS OMV groups on Day 10. However, by Day 20, there was a significant difference between the DSS and control groups (Fig. S5). There was a significant reduction in the number of bacterial species in the DSS group compared to that in the control group. However, in the DSS-treated OMV group, a greater diversity of bacterial species could be observed at the end of the experiment. The numerical bacterial diversity in the DSS-treated OMV group approached the initial diversity observed in the control group. The tables showing the relative abundance of bacterial species in each sample and statistics of the processed reads can be found in Table S3.

Analysis of the relative abundance of bacteria at the genus level allowed us to follow the change in bacterial composition from the control group to the DSS + OMV group. According to the data obtained, the relative abundance of the studied bacterial genera in the DSS + OMV group reached the initial values determined in the control group by the 20th day of observation. In contrast to the DSS group, where dramatic changes occurred, the number of microorganisms in the DSS + OMV group tended to return to control values when the vesicles were used (Fig. 5A). A Venn diagram and UpSet plot can be used to visualize the overlap in bacterial species between the three groups shown. The DSS + OMV group had a similar species composition to that of the control group (117 species). There were 30 common species between the DSS-treated OMV group and the DSS group. The DSS-treated OMV group had more than 100 bacterial species in common with the control group, whereas the DSS group and the control group had only 30 species in common (Fig. 5B and C).

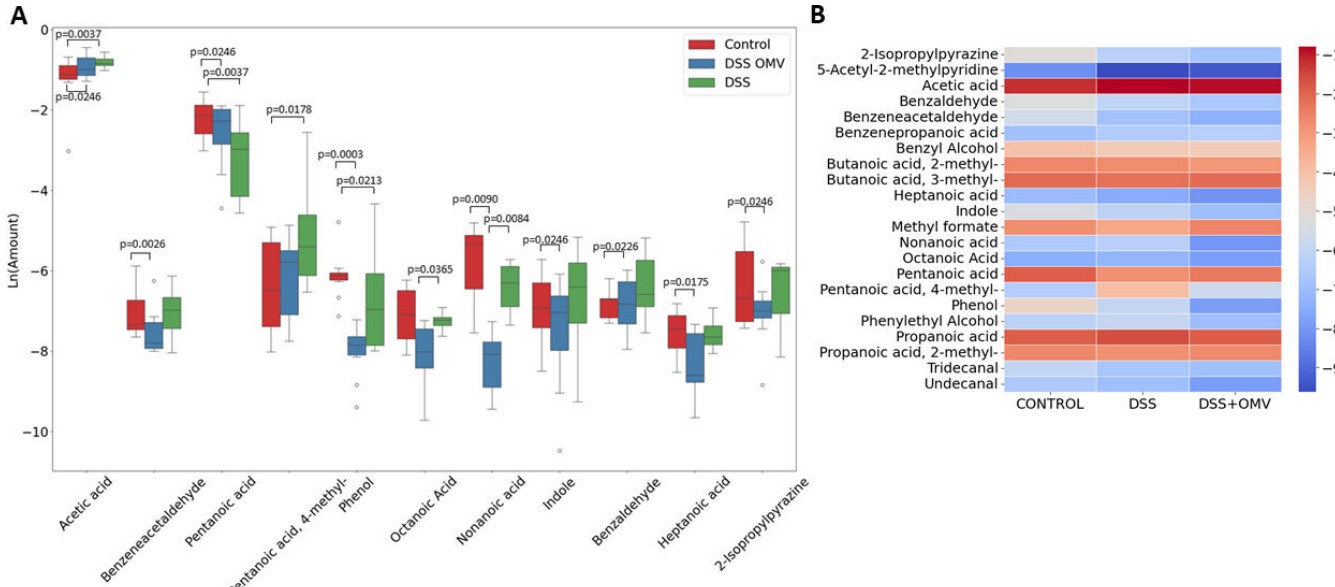

**FIG 3** Quantitative differences in the levels of individual components among the experimental groups. (A) Box plots showing the quantitative differences in the relative contents of short-chain fatty acids, amino acid derivatives, and others detected in the analyzed groups. The nonparametric Mann−Whitney test was used for the primary comparisons between groups. Statistical significance was determined by a two-sided *P* value of less than 0.05. FDR correction was also applied. (B) Comparison of VOC (volatile compound) composition on Day 20 in all groups. Relative concentrations in the vapor phase were used.

An alpha diversity study was performed to assess the homogeneity and richness of the microbial community. The results of the analysis showed that the alpha diversity values of the three groups were significantly different only on Day 20. Species diversity in the DSS group was significantly different from that in the control and vesicle-treated groups on Day 20 of the experiment. The assessment of alpha diversity revealed that all three groups were significantly different on the last day of the experiment, but the difference between the control group and the DSS group was more pronounced than the difference between the control group and the group receiving vesicles (DSS OMV) (Fig. 5D). Beta diversity analysis was used to quantify the differences between the bacterial communities of the samples. Principal coordinate analysis (PCoA) revealed that the bacterial communities were significantly different among all three groups (Fig. 5E). The Manhattan plot also showed significant differences among the three groups (Fig. S6). Linear discriminant analysis (LDA) was performed to search for significant bacterial taxa for each experimental group (27). The results showed that the OTUs were significantly enriched in the control group and belonged mainly to the phylum *Candidatus Saccharibacteria* and the genus *Acetivibrio*. The OTUs that were significantly enriched in the DSS group were mainly from the genera *Lactococcus* and *Romboutsia*, but the OTUs that were significantly enriched in the DSS + OMV group were mainly from the genus *Faecalibacterium* (Fig. 6).

Numerous significant correlations were found between microbiota composition and metabolomic compounds (Fig. 7). The alpha diversity indices and the relative abundance of *Saccharibacteria* and *Acetivibrio* were found to be positively correlated with phenol and pentanoic acid levels. In addition, as mentioned above, the alpha diversity indices and the abundance of *Saccharibacteria* and *Acetivibrio* were increased in the control group compared to the DSS and DSS + OMV groups (Fig. 5D and 6). The microbiota of DSS group is characterized by an increased representation of Lactococcus and Romboutsia (Fig. 6), and it was also found that their increase is associated with decreased phenol and pentanoic acid concentrations (Fig. 7). Thus, taking into account the observed associations between the composition of the microbial community and the studied

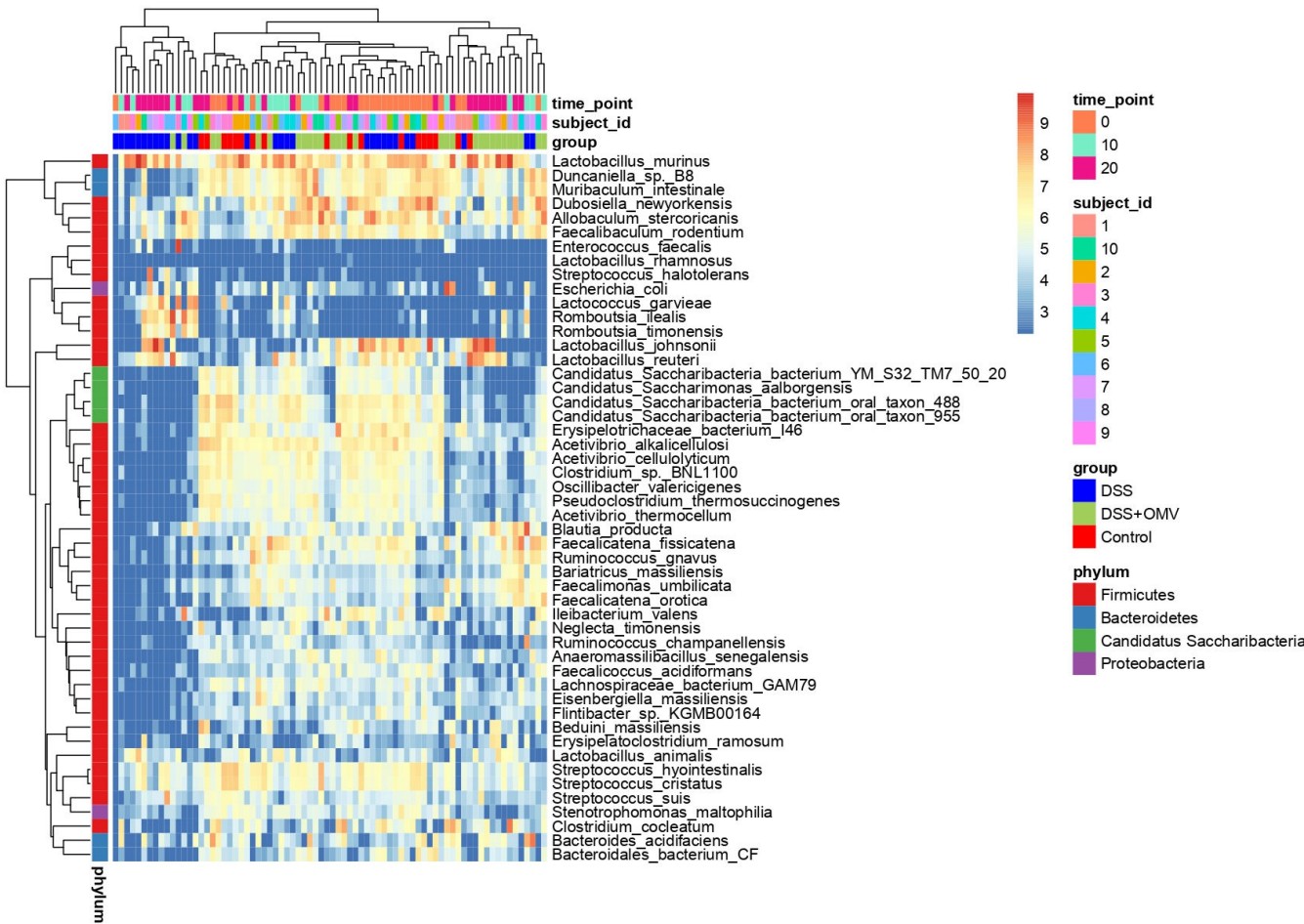

**FIG 4** Bacterial diversity at different time points (1, 10, and 20 days). The color scale of the heatmap shows the abundance of different species. The horizontal axis represents samples, and the vertical axis represents different bacterial species.

metabolites, a decrease of phenol and pentanoic acid levels may be a marker of dysbiosis associated with the DSS effects.

## DISCUSSION

DSS-induced colitis is a useful experimental model for studying the pathogenesis of inflammatory bowel disease and for testing new therapeutic agents (33, 34). DSS causes intestinal damage and triggers all stages of classical inflammation (35). The immune system is also involved in this process, which is of paramount importance in inflammatory bowel diseases such as Crohn's disease and ulcerative colitis due to the development of autoimmunity (36). Therefore, existing therapies are mainly aimed at suppressing the autoimmune response that provokes chronic intestinal inflammation (37). With the development of a chronic inflammatory process, the microbiota inevitably suffers (38). It has been shown that with the development of Crohn's disease and ulcerative colitis, a different ratio of bacterial species forms, as does the appearance of a greater number of pathogenic bacteria (39). These morphological changes lead to a decrease in the activity of the symbiotic microbiota, which has a negative effect on the gut (40). In addition, the normal microbiota contains unique species whose influence on the intestinal mucosa helps to reduce the inflammatory response (41). *B. fragilis* secretes vesicles with PSA on their surface (9). Several studies have shown that isolated PSA improves mucosal recovery in an induced colitis model (13). We suggest that the use of the OMVs for the treatment of IBD is more justified than the use of isolated PSA because

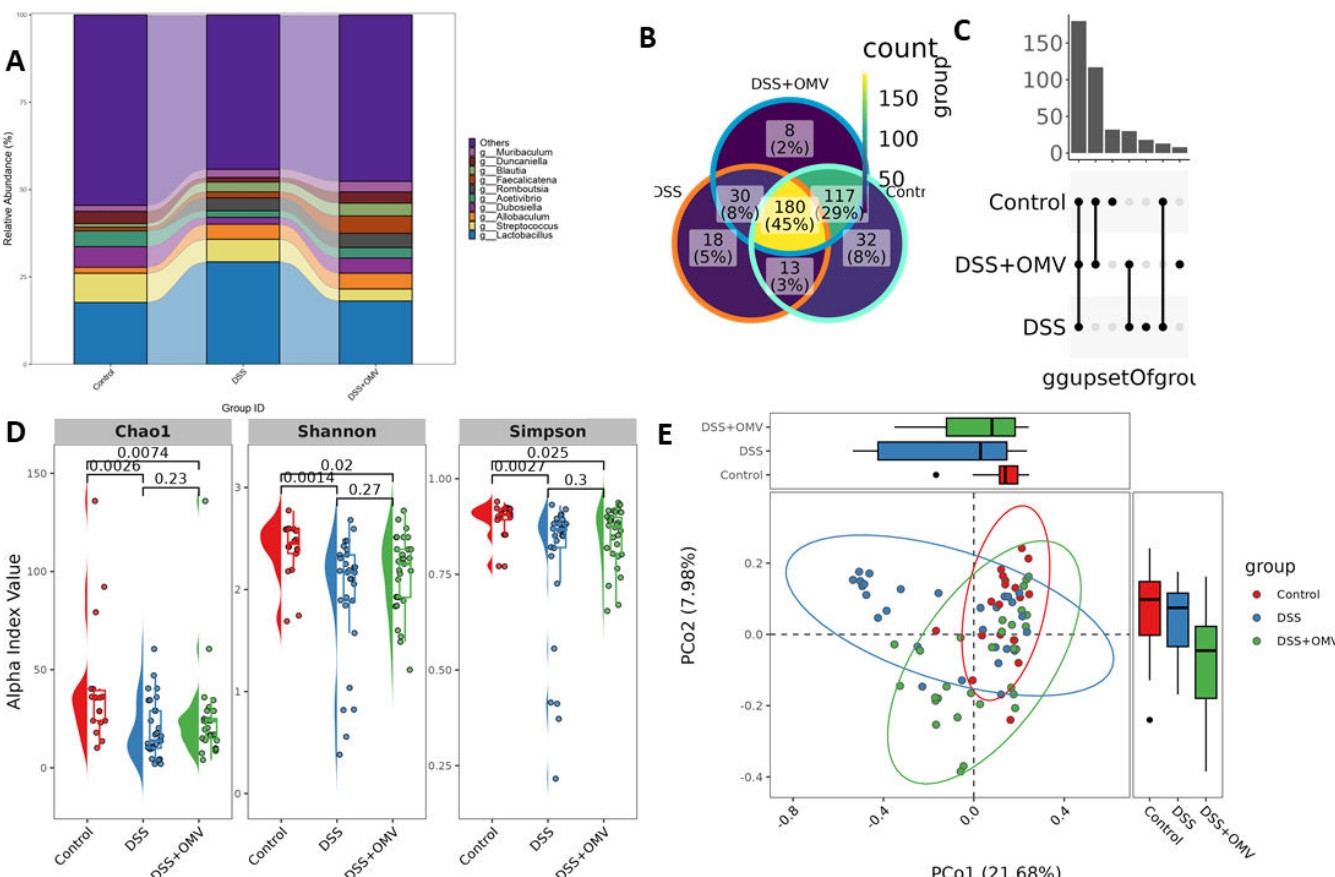

**FIG 5** Bacterial diversity at different time points (1, 10, and 20 days). (A) Relative abundance of bacteria at the genus level for each group (control, DSS, and DSS +OMV) across all sites. The horizontal axis shows groups, and the vertical axis shows relative abundance. (B, C) Venn diagram and UpSet plot for groups (control, DSS, and DSS + OMV) at the OTU level. (D) Plot of the alpha diversity index at all points. The horizontal axis represents each group (control, DSS, and DSS + OMV), and the vertical axis represents the alpha diversity index. (E) PCoA plot based on the Bray−Curtis distance for all time points. Each point represents one sample. The color of the point indicates the name of the sample group.

vesicles contain a significant number of enzymes that help to improve digestive function (14). In this study, we recreated a model of intestinal inflammation in the presence of DSS and treated it with *B. fragilis* JIM10 vesicles for 10 days. We first focused on the histological and physical assessment of the experimental animals. As expected, on Day 10 of DSS exposure, we observed changes in the intestinal tissue according to histopathology examination. However, as early as Day 20 of the experiment, when we used OMVs, we observed a positive therapeutic effect, which was confirmed by histological examination. Compared to the group without OMV treatment, the group that received vesicles was better able to repair the structure of the intestinal crypts and reduce the amount of inflammatory infiltrate in the tissue. In fact, we obtained an effect similar to that previously demonstrated with isolated PSA (15). When we analyzed the microbiota composition, we observed bacterial diversity in all three groups. For alpha and beta diversity, we observed changes in both the DSS and DSS-OMV groups. Importantly, when OMVs were used as therapy, the microbiota composition in the DSS-treated OMV group tended to return to its original control abundance. However, each group was characterized by its own unique bacterial diversity on Day 20. Our results showed that the Phylum *Candidatus Saccharibacteria* and the genus *Acetivibrio* were mainly enriched in the control group. *Lactococcus* and *Romboutsia* were significantly enriched in the DSS group, and the genus *Faecalibacterium* was significantly enriched in the DSS + OMV group. It can be assumed that the OMVs contributed to the recovery of the homeostasis of the original gut, which is a combination of the healthy mucosa layer and specific microbiota

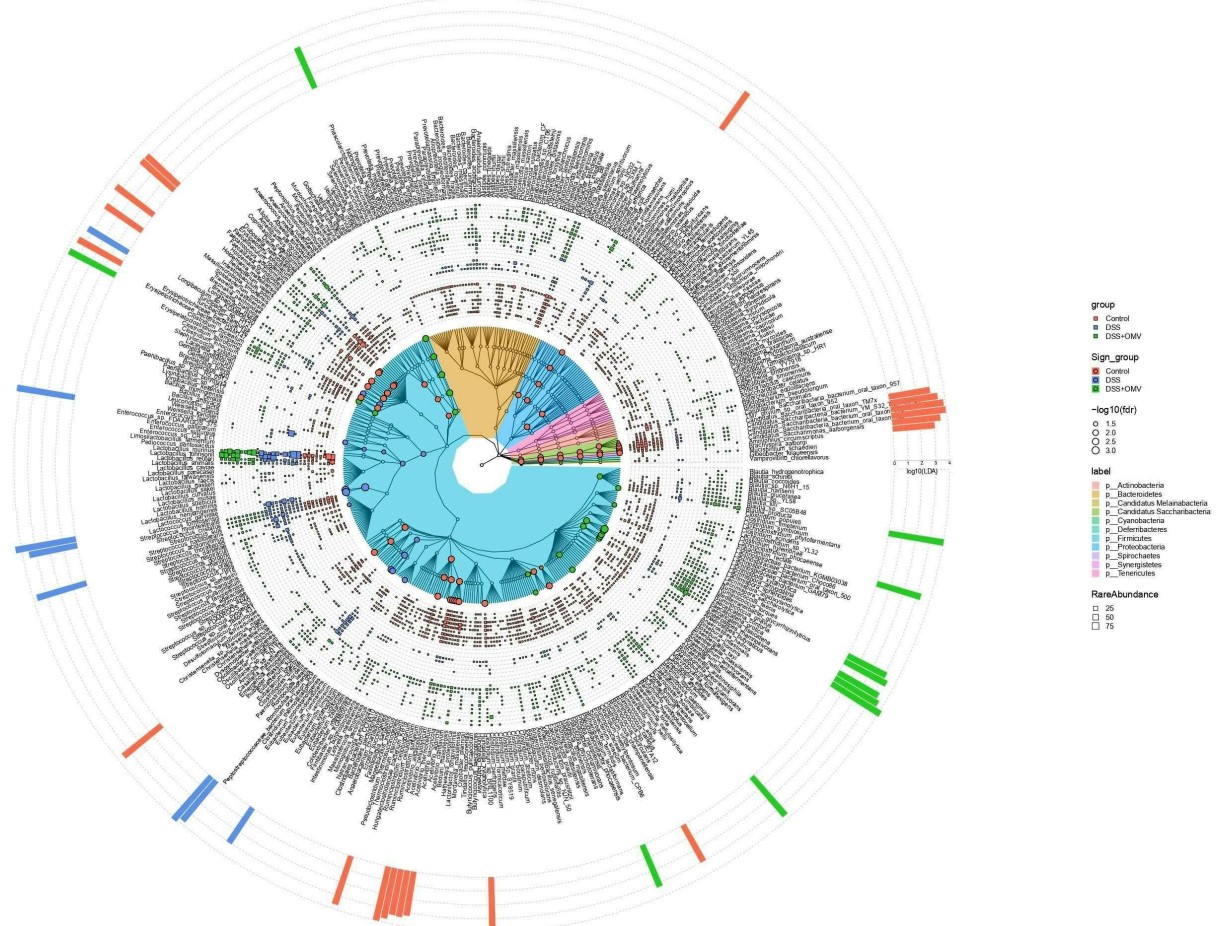

**FIG 6** Taxonomic community tree with the relative abundance of each OTU in the sample and the LDA of different OTUs. The tree was constructed using taxa from all samples. Colored clades represent phyla. The outer layer shows the relative abundance of each OTU in the sample. The outer histogram shows the LDAs of different OTUs. The colored dots represent different taxa, and the dot size indicates the FDR.

composition. Nevertheless, the changes in microbiota composition under DSS treatment were not as dramatic as might be expected. However, the influence of OMVs on microbiota composition is unique.

The metabolomic data complemented the 16S rRNA sequencing data. Changes in the relative amounts of volatile components, such as short-chain fatty acids, phenols, and pentanoic acid, were detected. Correlations of metabolites such as phenol and pentanoic acid with indices of microbiota diversity in the groups were observed. The relative amounts of these metabolites decreased in both experimental groups. *Saccharibacteria* and *Acetivibrio* were positively correlated with phenol and pentanoic acid in the control group, whereas *Lactococcus* and *Romboutsia* were negatively correlated with phenol and pentanoic acid in the DSS group. Based on the metabolomic and genomic data obtained, we can clearly conclude that DSS can induce changes in the microbiota composition, but *B. fragilis* OMVs contribute to the recovery of the original ratios.

We hypothesize that vesicles can directly influence the microbiota composition, delivering enzymes and metabolites necessary to restore microbiota functional activity. On the other hand, intestinal mucous layer repair under OMV treatment can also contribute to the recovery of the original microbiota composition.

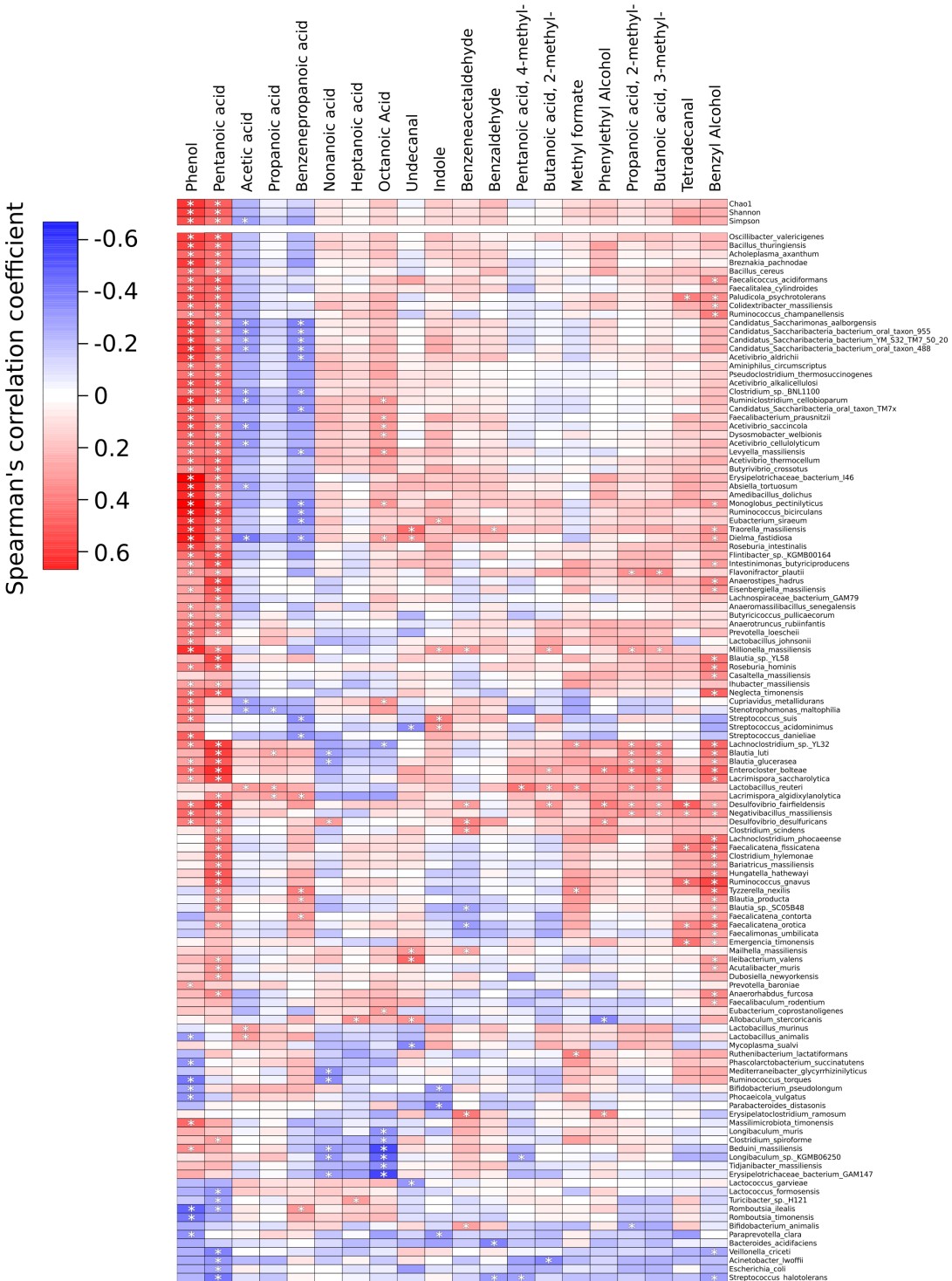

**FIG 7** Spearman's correlations between metabolite levels and relative abundances of microbial species. * *P* < 0.05.

## Conclusion

The aim of the present study was to evaluate the therapeutic effects of *B. fragilis* JIM10 OMVs on the recovery of DSS-affected intestinal tissue in a murine model. According to the data obtained, it was possible to demonstrate the therapeutic effectiveness of *B. fragilis* JIM10 OMVs and to observe the unique ability of OMVs to contribute to the recovery of the intestinal microbiota composition.

## ACKNOWLEDGMENTS

This research was performed using the core facilities of the Lopukhin FRCC PCM "Genomics, proteomics, metabolomics" (http://rcpcm.org/?p=2806). The authors thank the Center for Precision Genome Editing and Genetic Technologies for Biomedicine, Lopukhin Federal Research, and Clinical Center of Physical-Chemical Medicine of Federal Medical Biological Agency for providing computational resources for this project.

The TEM measurements were carried out at the User Facilities Center "Electron Microscopy in Life Sciences" of Lomonosov Moscow State University. The authors thank Mr. Alexey Senkovenko and Mr. Dmitry V. Bagrov for assistance with TEM imaging.

This research was supported by RSF grant 21-75-10172.

O.Y.S., D.A.K., V.A.I., E.A.V., M.I.M., E.I.O., and N.B.Z. designed and performed the experiments, analyzed the data, and wrote the paper. D.A.K., S.S.E., E.A.Z., O.V.A., Ya.A.Z., A.Y.M., A.S.S., B.A.E., A.A.V., M.D.M., O.Y.Z., and D.I.B. performed the experiments. T.V.G. and N.B.Z. supervised the project.

## AUTHOR AFFILIATIONS

[1]Lopukhin Federal Research and Clinical Center of Physical-Chemical Medicine of Federal Medical Biological Agency, Moscow, Russia

[2]The Laboratory of Ecological Genetics, Vavilov Institute of General Genetics, Russian Academy of Sciences, Moscow, Russia

[3]Department of Microbiology and Virology, Pirogov Russian National Research Medical University, Moscow, Russia

[4]Department of Basic and Applied Neurobiology, V. P. Serbsky National Medical Research Center for Psychiatry and Narcology, Moscow, Russia

[5]Institute of Fundamental Medicine and Biology, Kazan (Volga Region) Federal University, Kazan, Russia

[6]Vladimir Zelman Center for Neurobiology and Brain Rehabilitation, Skolkovo Institute of Science and Technology, Moscow, Russia

## AUTHOR ORCIDs

Evgenii I. Olekhnovich http://orcid.org/0000-0003-4899-342X
Natalya B. Zakharzhevskaya http://orcid.org/0000-0003-1045-1895

## FUNDING

| Funder | Grant(s) | Author(s) |
| --- | --- | --- |
| Russian Science Foundation (RSF) | 21-75-10172 | Natalya B. Zakharzhevskaya |

## AUTHOR CONTRIBUTIONS

Olga Yu. Shagaleeva, Conceptualization, Data curation, Formal analysis, Funding acquisition, Methodology, Validation, Visualization, Writing – original draft, Writing – review and editing | Daria A. Kashatnikova, Conceptualization, Methodology, Writing – original draft | Dmitry A. Kardonsky, Conceptualization, Formal analysis, Methodology, Writing – original draft | Boris A. Efimov, Conceptualization, Methodology, Writing – original draft | Viktor A. Ivanov, Methodology, Writing – original draft | Svetlana V. Smirnova, Data curation, Methodology | Suleiman S. Evsiev, Methodology | Eugene A. Zubkov, Methodology | Olga V. Abramova, Methodology | Yana A. Zorkina, Methodology | Anna Y. Morozova, Methodology | Elizaveta A. Vorobeva, Data curation, Formal analysis, Methodology, Validation, Writing – original draft | Artemiy S. Silantiev, Conceptualization, Methodology | Irina V. Kolesnikova, Conceptualization | Maria I. Markelova, Data curation, Formal analysis, Methodology, Writing – original draft | Evgenii I. Olekhnovich, Methodology, Writing – original draft | Maxim D. Morozov, Methodology | Polina Y. Zoruk, Methodology | Daria I. Boldyreva, Methodology | Victoriia D. Kazakova, Formal analysis | Anna A. Vanyushkina, Conceptualization | Andrei V. Chaplin, Conceptualization

| Tatiana V. Grigoryeva, Conceptualization | Natalya B. Zakharzhevskaya, Conceptualization, Data curation, Formal analysis, Funding acquisition, Investigation, Methodology, Project administration, Resources, Software, Supervision, Validation, Visualization, Writing – original draft, Writing – review and editing

## DATA AVAILABILITY

Raw metabolomic data are available on Mendeley.com, DOI: 10.17632/vfprtkk4cm.1. Raw metagenomic data are available with SubmissionID: SUB14689183; BioProject ID: PRJNA1152943.

## ETHICS APPROVAL

All experimental procedures were set up and maintained in accordance with Directive 2010/63/EU of 22 September 2010 and approved by the local ethical committee of V.P. Serbsky National Medical Research Center for Psychiatry and Narcology.

## ADDITIONAL FILES

The following material is available online.

### Supplemental Material

**Fig. S1 (Spectrum00636-24-s0001.tif).** TEM image of the isolated *Bacteroides fragilis* OMVs.
**Fig. S2 (Spectrum00636-24-s0002.pdf).** Histological examination of intestinal tissue from control group.
**Fig. S3 (Spectrum00636-24-s0003.pdf).** Histological examination of intestinal tissue from DSS group.
**Fig. S4 (Spectrum00636-24-s0004.pdf).** Histological examination of intestinal tissue from DSS+OMV group.
**Supplemental material (Spectrum00636-24-s0005.docx).** Fig. S5 and S6; legends.
**Table S1 (Spectrum00636-24-s0006.xlsx).** Histology and DAI indexes, obtained during intestine histology examination and animals observation.
**Table S2 (Spectrum00636-24-s0007.xls).** Metabolomic data obtained from control, DSS, and DSS+OMV groups.
**Table S3 (Spectrum00636-24-s0008.xlsx).** Sequencing data obtained from control, DSS, and DSS+OMV groups.

### Open Peer Review

**PEER REVIEW HISTORY (review-history.pdf).** An accounting of the reviewer comments and feedback.

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
