## [Reviewer comments · Microbiology Spectrum]

Microbiology Spectrum

***Bacteroides* vesicles promote functional alterations in the gut microbiota composition**

Olga Shagaleeva, Daria Kashatnikova, Dmitry Kardonsky, Boris Efimov, Viktor Ivanov, Svetlana Smirnova, Suleiman Evsiev, Eugene Zubkov, Olga Abramova, Yana Zorkina, Anna Morozova, Elizaveta Vorobeva, Artemiy Silantiev, Irina Kolesnikova, Maria Markelova, Evgenii Olekhovich, Maxim Morozov, Polina Zoruk, Daria Boldyreva, Victoria Kozakova, Anna Vanyushkina, Andrei Chaplin, Tatiana Grigoryeva, and Natalya Zakharzhevskaya

Corresponding Author(s): Natalya Zakharzhevskaya, Lopukhin Federal Research and Clinical Center of Physical-Chemical Medicine of Federal Medical Biological Agency, Moscow, Russian Federation

Review Timeline:

Submission Date:	March 10, 2024
Editorial Decision:	April 30, 2024
Revision Received:	June 27, 2024
Editorial Decision:	July 11, 2024
Revision Received:	August 6, 2024
Accepted:	August 9, 2024

Editor: Jan Claesen

Reviewer(s): Disclosure of reviewer identity is with reference to reviewer comments included in decision letter(s). The following individuals involved in review of your submission have agreed to reveal their identity: Aleksandar D Kostic (Reviewer #2)

Transaction Report:

DOI: <https://doi.org/10.1128/spectrum.00636-24>

Re: Spectrum00636-24 (Bacteroides vesicles promote functional alterations in gut microbiota composition)

Dear Dr. Natalya B. Zakharzhevskaya:

Thank you for the privilege of reviewing your work. Below you will find my comments, instructions from the Spectrum editorial office, and the reviewer comments.

Thanks for submitting your work to Spectrum. Your paper has been evaluated by an independent Reviewer, who is enthusiastic about your research (as am I), and has suggested some areas where the manuscript could be improved. I would be happy to consider a revised version of your paper addressing these Reviewer comments in a point-by-point manner. Several of the comments pertain to the claims not being fully supported by the data, hence it might be recommended to rephrase some of these claims to obtain a more balanced narrative.

Note that for a resubmission, Spectrum requests a data availability statement, which should be included in line with the author guidelines.

While we are willing to consider a revised version of this paper at Spectrum, it would be in your best interests to improve the writing. I recommend that you ask a colleague of yours whose native English speaker to read and provide you some feedback on the writing. You are also welcome to use one of the services here:
<https://journals.asm.org/content/language-editing-services>

Revision Guidelines

Sincerely,

Reviewer #1 (Comments for the Author):

The manuscript "Bacteroides vesicles promote functional alterations in gut microbiota composition" by Olga Shagaleeva et al., expands on an in vivo chronic model of DSS-induced intestinal damage and show modulation of microbiota content at the taxonomic and metabolic level.

Additionally, the authors explore the protective role of outer membrane vesicles (OMV) from *Bacteroides fragilis*, assessed by disease activity index and histology, as well as by taxonomic and metabolic criteria.

While I find the manuscript to be interesting and relevant to the field of gut microbiota, as presented, the manuscript has several items that should be addressed to support the main claims.

Major points:

- The image data provided in Figure 1 is not sufficient to support such an extensive description of the histological features for the different groups. The authors should provide more representative images that better describe their findings, or limit their descriptions.
- Alcian Blue staining of Goblet cells is unclear, the authors should provide higher quality stain images and a proper quantification, or remove goblet cell description from the manuscript.
- Statistical significance and p-values must be indicated in Figure 1B and 1C
- The figures, even when downloaded as separate, must have a larger letter size for the labels of the charts, they become too difficult to read as they are. Figure 3A labels are too small to interpret the figure correctly, and asterisks and p values should be included as well.
- Genus *Hungateiclostridium* should be revisited by the authors and modified to *Acetivibrio*, based on nomenclature criteria (<https://www.microbiologyresearch.org/content/journal/ijsem/10.1099/ijsem.0.003685>).

Minor points:

- The manuscript could benefit from a general revision of grammar content to improve clarity.
- Labels on table S2 are not clear in their meaning, and multiple compound names are preceded by question marks. Authors should improve the labelling and symbols of the table.
- Bacterial nomenclature must be more precise and in italics when referring to genus and species. Examples below:
- Lines 34/35 and 51, correct spelling should be *Hungateiclostridium*.
- Lines 41 and 43, should say *Bacteroides fragilis*.
- Lines 371/2, Phylum *Candidatus Saccharibacteria* and the genus *Hungateiclostridium*.
- Line 336, should say Firmicutes. Phyla names start with a capital letter but non-italicized.
- Phrase on lines 328/329 should be removed, the authors cannot sustain that claim with the presented data.
- Figure 7 data analysis in lines 375 through 381 is difficult to follow. The authors should link the taxonomic information from Figures 5/6 with the chart from Figure 7 using a more clear and concise phrasing.

The manuscript “*Bacteroides* vesicles promote functional alterations in gut microbiota composition” by Olga Shagaleeva et al., expands on an *in vivo* chronic model of DSS-induced intestinal damage and show modulation of microbiota content at the taxonomic and metabolic level.

Additionally, the authors explore the protective role of outer membrane vesicles (OMV) from *Bacteroides fragilis*, assessed by disease activity index and histology, as well as by taxonomic and metabolic criteria.

While I find the manuscript to be interesting and relevant to the field of gut microbiota, as presented, the manuscript has several items that should be addressed to support the main claims.

Major points:

- The image data provided in Figure 1 is not sufficient to support such an extensive description of the histologic features for the different groups. The authors should provide more representative images that better describe their findings, or limit their descriptions.
- Alcian Blue staining of Goblet cells is unclear, the authors should provide higher quality stain images and a proper quantification, or remove goblet cell description from the manuscript.
- Statistical significance and p-values must be indicated in Figure 1B and 1C
- The figures, even when downloaded as separate, must have a larger letter size for the labels of the charts, they become too difficult to read as they are. Figure 3A labels are too small to interpret the figure correctly, and asterisks and p values should be included as well.
- Genus *Hungateiclostridium* should be revisited by the authors and modified to *Acetivibrio*, based on nomenclature criteria (<https://www.microbiologyresearch.org/content/journal/ijsem/10.1099/ijsem.0.003685>).

Minor points:

- The manuscript could benefit from a general revision of grammar content to improve clarity.
- Labels on table S2 are not clear in their meaning, and multiple compound names are preceded by question marks. Authors should improve the labelling and symbols of the table.
- Bacterial nomenclature must be more precise and in italics when referring to genus and species. Examples below:
- Lines 34/35 and 51, correct spelling should be *Hungateiclostridium*.
- Lines 41 and 43, should say *Bacteroides fragilis*.
- Lines 371/2, Phylum Candidatus Saccharibacteria and the genus *Hungateiclostridium*.
- Line 336, should say Firmicutes. Phyla names start with a capital letter but non-italicized.
- Phrase on lines 328/329 should be removed, the authors cannot sustain that claim with the presented data.

- Figure 7 data analysis in lines 375 through 381 is difficult to follow. The authors should link the taxonomic information from Figures 5/6 with the chart from Figure 7 using a more clear and concise phrasing.

Dear college

We are very thankful for the comprehensive analysis of our research. Our detailed answers are listed below in italic.

Major points:

1. The image data provided in Figure 1 is not sufficient to support such an extensive description of the histologic features for the different groups. The authors should provide more representative images that better describe their findings, or limit their descriptions.

We obtained comprehensive histological data in our experiment, which we would like to describe in detail in the manuscript. We present additional images of histological sections of intestine from the control group (Fig S1), DSS group (Fig S2), and DSS OMV group (Fig S3) in supplementary material. We also provide information about the appearance of additional histological data in the manuscript: P9 line 254, P9 line 256, P9 line 263

2. Alcian Blue staining of Goblet cells is unclear, the authors should provide higher quality stain images and a proper quantification, or remove goblet cell description from the manuscript

We have thought through this issue in detail. The fact is that counting goblet cells in the mucous membrane of control animals is not difficult. However, difficulties arise with quantitative counting of goblet cells in experimental groups. Counting cells near the site of destruction of the mucous membrane is not indicative, since when normalizing for the length of the mucous membrane, swelling of the mucous membrane and a decrease in high crypts are not taken into account. Thus, we came to the conclusion that quantitative presentation of data is impossible due to the lack of standardization of counts between animals of different groups. In the text we additionally indicated that in the control group, after staining, goblet cells are well visualized, however, in the experimental group, due to significant destruction of the mucous membrane, we cannot accurately count the number of goblet cells.

P9 line 265-267

3. Statistical significance and p-values must be indicated in Figure 1B and 1C

The required data has been entered in Figure 1B and 1C

P20 line 596-597

4. The figures, even when downloaded as separate, must have a larger letter size for the labels of the charts, they become too difficult to read as they are. Figure 3A labels are too small to interpret the figure correctly, and asterisks and p values should be included as well.

We have increased the fonts in all Figures including 3A; p values were added.

P21 line 611

5. Genus *Hungateiclostridium* should be revisited by the authors and modified to *Acetivibrio*, based on nomenclature criteria

Thank you very much for this important note. We have made the necessary corrections (in the manuscript and Figures) according to the latest data on the nomenclature of this genus.

Minor points:

1. The manuscript could benefit from a general revision of grammar content to improve clarity

The manuscript was additionally checked by a native speaker

2. Labels on table S2 are not clear in their meaning, and multiple compound names are preceded by question marks. Authors should improve the labelling and symbols of the table.

Necessary changes have been made

3. Bacterial nomenclature must be more precise and in italics when referring to genus and species. Examples below:

- Lines 34/35 and 51, correct spelling should be *Hungateiclostridium*.

We renamed Hungateiclostridium to Acetivibrio according the last nomenclature criteria

- Lines 41 and 43, should say *Bacteroides fragilis*.

Necessary changes have been made

- Lines 371/2, Phylum Candidatus Saccharibacteria and the genus *Hungateiclostridium*.

Necessary changes have been made

- Line 336, should say Firmicutes. Phyla names start with a capital letter but non-italicized.

Necessary changes have been made

- Phrase on lines 328/329 should be removed, the authors cannot sustain that claim with the presented data.

Necessary changes have been made

Figure 7 data analysis in lines 375 through 381 is difficult to follow. The authors should link the taxonomic information from Figures 5/6 with the chart from Figure 7 using a more clear and concise phrasing

We performed a new description of obtained data:

Different correlations between microbiota composition and metabolomic compounds were found (Figure 7). It was found that the increase alpha diversity community associated with the control group (Fig. 5D) was positively correlated with the relevant content of phenol and pentanoic acid. In addition, positive correlations of these metabolites with the representation of Saccharibacteria and Acetivibrio were found, which were also increased in the control group (Fig. 4). Lactococcus and Romboutsia increase was the main characteristic of the DSS group (Fig. 4) and it was additionally revealed that its increase is associated with a decrease in the content of phenol and pentanoic acid (Fig. 7). Thus, a decrease in the content of phenol and pentanoic acid may be a marker of dysbiosis associated with the action of DSS.

P13 line 371-378

Re: Spectrum00636-24R1 (Bacteroides vesicles promote functional alterations in gut microbiota composition)

Dear Dr. Natalya B. Zakharzhevskaya:

Thank you for the privilege of reviewing your work. Below you will find my comments, instructions from the Spectrum editorial office, and the reviewer comments.

Thank you for carefully addressing Reviewer 1's comments! Some time between me sending my previous decision letter and receipt of your revised manuscript, the second Reviewer did come through and they provided a thorough evaluation.

These are unusual circumstances and I apologize for the back and forth process, but I would like to ask if you could also address Reviewer 2's comments in a revised manuscript, as they will help improve the paper.

Reviewer 2 comments are pasted below:

Summary of findings and claims:

In this study, Shagaleeva and colleagues present the findings of a 10-day treatment on DSS-induced colitis mice using *Bacteroides fragilis* (strain JIM10) OMVs. The authors observed that the OMV-treated group exhibited improved intestinal crypt structure repair and reduced tissue inflammation, mirroring effects seen with isolated PSA. They report that DSS-OMV group showed a tendency to restore its initial microbiota composition in terms of relative abundance, closely resembling the control group. They argue that by the end of the experiment, different bacterial diversities were identified in each group: *Candidatus saccharibacteria* and *Hungatei* in the control group, *Lactococcus* and *Romboutsia* in the DSS group, and *Faecalimonas* in the DSS+OMV group. The authors suggests that *Bacteroides* OMVs support gut homeostasis recovery. The authors also found *Saccharibacteria* and *Hungatei* clostridium positively correlated with phenol and pentanoic acid in the control group, whereas *Lactococcus* and *Romboutsia* negatively correlated in the DSS group. The study concludes that *Bacteroides fragilis* OMVs help restore tissue damage and the original microbial composition in the DSS-induced colitis model.

Major Concerns:

It's not clear that the severity of colitis in the DSS+OMV group is significantly different from the DSS group. The statistical tests used are not in the figure captions, and I was unable to find them in the main text either. This is a critical result to the rest of the manuscript, but it is not sufficiently justified. As the authors indicate, Mazmanian previously reported that *B.frag* OMVs can prevent colitis in TNBS model, so this would not be a novel result, but it is still critical to the remaining claims of the paper. As in Fig.1, in the rest of the manuscript there is almost no clear, convincing result that can serve as a handle for defending any major claims in this manuscript. It is difficult to know where to begin with the problems because there are so many, but I have tried to do my best to be as comprehensive as possible below for the future benefit of the authors.

The figure legend and most of the associated text is COMPLETELY unlegible, making any critique of the manuscript extremely time consuming.

Page 1 line 30: The authors claim that the comparative metabolomic analysis suggested a change in functional activity between the DSS and DSS OMV groups. However, the paper doesn't provide clear evidence to substantiate this functional claim; a relative comparison of "volatile compounds" between the groups is insufficient to make a functional conclusion.

Throughout the paper, the authors interchange the genus term *Bacteroides* with the species level *Bacteroides fragilis*, and the strain level *Bacteroides fragilis* JIM10. This results in confusing statements. A prime example is the title of the manuscript, "*Bacteroides* vesicles promote functional alterations in gut microbiota composition."; the study can and should only make claims at the species level at most.

Many sources cited in the manuscript's introduction incorrectly reference the findings of the original paper. Furthermore, some claims lack supporting citations. Here are some examples:

Page 2 lines 27-28: IBD therapy does not primarily aim to reduce inflammation with hormones and antibiotics. Perhaps here they are referring to corticosteroids, but this is generally for short-term use and not a mainstay of IBD therapy.

Page 3 lines 1-2: Although the cited source Pobeguts et. al 2020 is relevant for the virulence properties (*Escherichia coli** colonizing macrophage) developed in CD isolate *E. coli** under propionate-rich media, no where in the source provide general evidence for Bacteria developing pathogenic phenotype to inflammatory intestinal environment.

Page 3, lines 10-11: I remain unconvinced by the authors' reasoning that *Bacteroides** species cannot be used as a single component of probiotics, even after reviewing the original source, Sun et al., 2019.

Page 3, lines 18-20: the authors state, "multi-component OMVs are expected to be more effective than isolated PSA in reducing inflammation and subsequent microbial colonization." However, the cited source, Mazmanian et al., 2008, provides no evidence supporting this; the source only discusses the effect of *B. fragilis* isolated PSA on colonic inflammation, with no mention of the complementary effect of OMVs. Mazmanian papers on OMVs were published in 2012 and later.

Page 3 lines 24-27: there's a lack of citation.

Page 3 lines 27-32: the manuscript states that "... reproduced a severe DSS response when introduced to germ-free mice, thereby confirming the causal relationship between these species and the severity of DSS colitis." However, the cited source, Forster et al. 2022, only suggests a "potential" cause for DSS endpoint variability. The observed difference in germ-free mice monocolonized endpoints is insufficient to validate a causal conclusion. Also, it is important to mention that the source's identified species are mouse specific.

Page 3 lines 32-33: there is a lack of citation for why significant changes in microbiota composition are not expected from a DSS model.

Page 4 lines 20-24 (OMV isolation): The authors' initial filtration step uses a 0.45 μm membrane, which is adequate for removing cells but may not efficiently remove all cell debris, which could contaminate the OMV preparation. As seen in Supplementary Figure S1, the OMV yield appears low, and potential contaminants in the OMV preparation could significantly undermine the paper's claim. A secondary filtration step could be considered prior to ultracentrifugation using a smaller pore size filter, such as 0.22 μm , to ensure more rigorous clearance of finer particulate matter. Also, vesicle pellet was resuspended in distilled water or 150 mM NaCl, however, using isotonic solutions that mimic physiological conditions (e.g., phosphate-buffered saline) could help maintain vesicle integrity better than distilled water.

Page 4 lines 25 (OMV isolation): The method mentions quantifying OMVs by dry precipitation but does not provide details on how this is carried out or why it was chosen. Dry weight can be a useful measure of total biomass but may not accurately reflect vesicle concentration or integrity. More commonly, protein content or nanoparticle tracking analysis might offer more detailed insights into vesicle quantity and size distribution.

Page 5 lines 13-14 (Animals): The methods section mentions an alternating schedule between DSS and water during the 20-day maintenance phase for the DSS and DSS+OMV groups. However, it lacks precise details about the DSS administration schedule, such as frequency and duration of DSS exposure. This lack of specificity could lead to variability in the degree of induced inflammation between subjects, thereby affecting the consistency of the experimental conditions.

Page 5, lines 15-17 (Animals): The manuscript fails to provide information about the frequency of OMV administration (e.g., daily, every other day). Additionally, it does not specify the route of OMV administration, which is crucial because the route (oral, intravenous, etc.) can significantly influence the therapeutic efficacy of OMVs. It would be essential to ensure that the volume and method of administration match exactly those of the treatment groups to maintain consistent handling stress across all groups.

Page 9, lines 8-9, and Figure 1A K3: the authors claim that "several animals from the control group were characterized by non-extended areas of goblet cell loss". However, it's unclear how these 3 histology slides were selected from the 10 samples per group. Additionally, there are no figures or data provided for readers to assess the "several" cases, with only 1 case being presented per group (K3, DSS3, DSSOMV-3).

Page 10, lines 9-10: The statement regarding the re-introduction of DSS is unclear because the methodology does not explicitly mention this.

Page 10, lines 16-17: (referencing Table 1 and Figure 2), the text mentions, "at least 60% of the detected stable compounds were used for the metabolomic profiles comparison." However, the significance of the 60% is unclear, and the inclusion/exclusion criteria for Table 1 and Figure 3 further compound the confusion. Are all figures from Figure 2 onwards solely based on the metabolite sample size of $n=263$ as mentioned in Table 1? Why does the PCA and OPLS DA data display only 8 samples per group, in contrast to the initial sample size per group of 10? Furthermore, why does the heatmap scale range from -1 to -12? It appears the heatmap is missing a label, which I assume indicates log-fold changes. If that's the case, the authors should consider implementing a better normalization strategy, improving the color scale, or providing a more detailed description in either explaining the results or in the methods section. The blank sections of the heatmap create confusion; it's unclear if these values are NAs or if they represent a log fold value of approximately -6.5. The methodology for the correlation analysis is unclear, and was not clear in delivery how to interpret the meaning of the paper's findings.

Page 11, lines 16-18: Authors mention significant depletion of the metabolic profile in the DSS group compared to the control and DSS OMV groups however, figure 2C is not enough to support this claim. There needs to be a supplementary figure or table or some numbers to support this. Also, the claim around the OMV's tendency to "restore the metabolite profile is also weak.

Page 11, lines 23-27: the paper states that "most of the significant differences were found between the control and DSS OMV groups" regarding metabolites. However, Figure 3C appears to contradict this statement looking carefully at that the figure legend (heatmap scale). Furthermore, the statement "It can be assumed that a new equilibrium of the relative amounts of individual metabolites can be observed after OMVs treatment" makes a strong assumption without substantial evidence.

Page 13, lines 12-18: I am unsure how the authors can draw comparative inferences from Figure 7, such as the "predominance of" x bacteria being negatively or positively "correlated with" metabolite y when comparing group A and group B. This concern is significant, as this figure is supposedly one of the main supports for the paper's claim.

Page 14, lines 2-4: The claim that the use of OMVs is more beneficial than isolated PSA due to the presence of enzymes improving digestive function lacks strong evidence. We need to specify what "digestive function" means and provide specific proof that the enzymes in OMVs directly contribute to better results in IBD treatment beyond delivering PSA.

Page 14, lines 20-22: The discussion implies that OMVs can aid in restoring the original gut homeostasis, but lacks substantial experimental evidence to establish a clear causal link between OMV treatment and specific changes in microbiota composition and gut mucosal healing. This claim should be supported by more detailed data or mechanistic insights. While the results highlight differences between the experimental groups, they don't provide sufficient evidence to support the "recovery" claim.

Additional Concerns:

Page 4 lines 14/16 (Bacterial strain and growth conditions): The protocol mentions cultivation until stationary phase but does not specify how this phase is determined. For experimental reproducibility, it is crucial to define the stationary phase more clearly, perhaps through optical density measurements at a specific wavelength (e.g., OD600) or via viable cell counts.

Page 5 line 14-19: While the paper includes an ethics statement, the use of chloroform for euthanasia is still concerning. Chloroform can cause respiratory distress and inconsistent times to loss of consciousness, potentially leading to unnecessary suffering and stress. This could also mildly impact experimental outcomes, especially those involving inflammatory processes. Current guidelines and standards recommend using methods like CO₂ asphyxiation followed by cervical dislocation.

Page 7 15-18 (Metabolome data processing): Assuming that the raw data do not follow a normal distribution without conducting any formal test for normality, such as the Shapiro-Wilk test, could be an issue. Although it's practical to use non-parametric tests like the Mann-Whitney test for non-normal distributions, it's vital to verify the type of distribution through appropriate tests. The methodology discussed the use of both the Mann-Whitney test and ANOVA after a natural log transformation. It would be helpful to provide a clearer rationale for the choice of each test based on the data distribution and to apply corrections for multiple comparisons.

Figure 1B/C: The plots representing histological index and DAI of the distal intestine show extremely wide confidence intervals with significant overlap between groups. A statistical comparison of mean values is needed, along with the calculation of p-values.

Page 11, lines 20-23: It is unclear how the compounds in the table shown in Figure 3B were selected.

Page 13, lines 5-11: Figure 6 has low resolution and cannot be interpreted.

Minor Concerns

Page 2 line 5: Could the authors define what they mean by "intermediate" in the phrase "microbiota composition of the DSS OMV group was intermediate between the control and DSS groups"?

Page 2 lines 9/11: I presume "Bacteroides fragilis" is a typo for *Bacteroides fragilis*.

Page 8 line 23: Versions for Porechop and NanoFilt softwares are missing.

Figure 3A: Labels unclear.

The authors should define "VOCs," which I assume stands for Volatile Organic Compounds, somewhere in the paper.

--- end of Reviewer 2 comments

Revision Guidelines

Sincerely,
Jan Claesen
Editor
Microbiology Spectrum

Dear college

We are very thankful for the comprehensive analysis of our research. Our detailed answers are listed below in italic.

Major points:

1. The image data provided in Figure 1 is not sufficient to support such an extensive description of the histologic features for the different groups. The authors should provide more representative images that better describe their findings, or limit their descriptions.

We obtained comprehensive histological data in our experiment, which we would like to describe in detail in the manuscript. We present additional images of histological sections of intestine from the control group (Fig S1), DSS group (Fig S2), and DSS OMV group (Fig S3) in supplementary material. We also provide information about the appearance of additional histological data in the manuscript: P9 line 254, P9 line 256, P9 line 263

2. Alcian Blue staining of Goblet cells is unclear, the authors should provide higher quality stain images and a proper quantification, or remove goblet cell description from the manuscript

We have thought through this issue in detail. The fact is that counting goblet cells in the mucous membrane of control animals is not difficult. However, difficulties arise with quantitative counting of goblet cells in experimental groups. Counting cells near the site of destruction of the mucous membrane is not indicative, since when normalizing for the length of the mucous membrane, swelling of the mucous membrane and a decrease in high crypts are not taken into account. Thus, we came to the conclusion that quantitative presentation of data is impossible due to the lack of standardization of counts between animals of different groups. In the text we additionally indicated that in the control group, after staining, goblet cells are well visualized, however, in the experimental group, due to significant destruction of the mucous membrane, we cannot accurately count the number of goblet cells.

P9 line 265-267

3. Statistical significance and p-values must be indicated in Figure 1B and 1C

The required data has been entered in Figure 1B and 1C

P20 line 596-597

4. The figures, even when downloaded as separate, must have a larger letter size for the labels of the charts, they become too difficult to read as they are. Figure 3A labels are too small to interpret the figure correctly, and asterisks and p values should be included as well.

We have increased the fonts in all Figures including 3A; p values were added.

P21 line 611

5. Genus *Hungateiclostridium* should be revisited by the authors and modified to *Acetivibrio*, based on nomenclature criteria

Thank you very much for this important note. We have made the necessary corrections (in the manuscript and Figures) according to the latest data on the nomenclature of this genus.

Minor points:

1. The manuscript could benefit from a general revision of grammar content to improve clarity

The manuscript was additionally checked by a native speaker

2. Labels on table S2 are not clear in their meaning, and multiple compound names are preceded by question marks. Authors should improve the labelling and symbols of the table.

Necessary changes have been made

3. Bacterial nomenclature must be more precise and in italics when referring to genus and species. Examples below:

- Lines 34/35 and 51, correct spelling should be *Hungateiclostridium*.

We renamed Hungateiclostridium to Acetivibrio according the last nomenclature criteria

- Lines 41 and 43, should say *Bacteroides fragilis*.

Necessary changes have been made

- Lines 371/2, Phylum Candidatus Saccharibacteria and the genus *Hungateiclostridium*.

Necessary changes have been made

- Line 336, should say Firmicutes. Phyla names start with a capital letter but non-italicized.

Necessary changes have been made

- Phrase on lines 328/329 should be removed, the authors cannot sustain that claim with the presented data.

Necessary changes have been made

Figure 7 data analysis in lines 375 through 381 is difficult to follow. The authors should link the taxonomic information from Figures 5/6 with the chart from Figure 7 using a more clear and concise phrasing

We performed a new description of obtained data:

Different correlations between microbiota composition and metabolomic compounds were found (Figure 7). It was found that the increase alpha diversity community associated with the control group (Fig. 5D) was positively correlated with the relevant content of phenol and pentanoic acid. In addition, positive correlations of these metabolites with the representation of Saccharibacteria and Acetivibrio were found, which were also increased in the control group (Fig. 4). Lactococcus and Romboutsia increase was the main characteristic of the DSS group (Fig. 4) and it was additionally revealed that its increase is associated with a decrease in the content of phenol and pentanoic acid (Fig. 7). Thus, a decrease in the content of phenol and pentanoic acid may be a marker of dysbiosis associated with the action of DSS.

P13 line 371-378

Dear colleague,

We are very thankful for the comprehensive analysis of our research. Our detailed answers are listed below in the table.

Questions	Answers	Line (for pdf version)
It's not clear that the severity of colitis in the DSS+OMV group is significantly different from the DSS group. The statistical tests used are not in the figure captions, and I was unable to find them in the main text either. This is a critical result to the rest of the manuscript, but it is not sufficiently justified.	We have included a detailed description of the statistical methods used in the research in material and methods section and in figure captions.	Line 163 Line 177 All Figure legends
As the authors indicate, Mazmanian previously reported that B.frag OMVs can prevent colitis in TNBS model, so this would not be a novel result, but it is still critical to the remaining claims of the paper. As in Fig.1, in the rest of the manuscript there is almost no clear, convincing result that can serve as a handle for defending any major claims in this manuscript. It is difficult to know where to begin with the problems because there are so many, but I have tried to do my best to be as comprehensive as possible below for the future benefit of the authors.	Dear colleague, we are very sorry that the narrative in the article turned out to be illogical and unclear for you. In our work, we relied on the results of colleagues, according to which vesicles have a therapeutic effect on the intestinal mucosa, and showed this on the DSS model, both in rats and in mice. The TNBS model causes transmural colitis, which is caused by a Th1-mediated immune response. In the same way, colitis is induced by oxazolone suspended in ethanol, which, unlike TNBS-induced colitis, promotes a Th-2-type cytokine response, with superficial lesions of the mucosa of the distal colon. However, the DSS model is most often used in modeling colitis in rodents. Unlike the other two models, it is well suited for studying innate immunity and inflammation and healing processes, due to good reproducibility, ease of dose modification and treatment/induction cycles. DSS causes damage to the colonic epithelium in all its sections. Thus, we reproduced the results of colleagues but on the DSS colitis model in animals. Our main objective in this study was to evaluate the microbiota composition and volatile metabolome of stool samples from experimental animals during OMV therapy. We took Bacteroides fragilis vesicles with the expected therapeutic effect, but in a DSS-induced colitis model and used the headspace extraction method	

	coupled with GC-MS/MS, which is rarely used for animals' stools analysis. It was interesting to see if the course of therapy can be monitored by analyzing of the microbiota composition by 16s RNA sequencing and volatile metabolome by HS GC-MS/MS. We would like to further apply the combination of these two methods for the early diagnosis of inflammatory bowel diseases in humans. In our recently published study (doi: 10.3390/ijms25063295) in a rat model of DSS-induced colitis, we showed that the headspace extraction method is highly sensitive and allows recording metabolome changes both in acute colitis and during spontaneous intestinal recovery. In this study, we test the possibility of using a combination of methods in the course of non-invasive assessment of therapy effectiveness. Therefore, we really hope that our explanation will help to look more optimistically at our results.	
The figure legend and most of the associated text is COMPLETELY unlegible, making any critique of the manuscript extremely time consuming.	We have modified all figures legends to make the analysis of the manuscript more convenient	
Page 1 line 30: The authors claim that the comparative metabolomic analysis suggested a change in functional activity between the DSS and DSS OMV groups. However, the paper doesn't provide clear evidence to substantiate this functional claim; a relative comparison of "volatile compounds" between the groups is insufficient to make a functional conclusion.	Today, the basis of functional metabolomics is formed by metagenomic analysis data obtained by sequencing of the microbiome genome and analysis of the metabolome of the volatile fraction of stool metabolites mainly produced by bacteria. In this regard, most of the published research based on the results obtained from a combined analysis of the composition of the microbiota and the secretory activity of intestinal microorganisms. In our work, we conduct a correlation analysis, linking the secretory activity of bacteria and the composition of the microbiota. Of course, we make an indirect conclusion that the change in the level of metabolites is a reflection of the changed functional activity of bacteria. However, by the functional activity of the microbiota we mean a change in the ratio of bacterial species, whose total secretory activity is assessed.	
Throughout the paper, the authors interchange the genus	We have reviewed the text and made the necessary edits. However, we	Line 67

term Bacteroides with the species level Bacteroides fragilis, and the strain level Bacteroides fragilis JIM10. This results in confusing statements. A prime example is the title of the manuscript, "Bacteroides vesicles promote functional alterations in gut microbiota composition."; the study can and should only make claims at the species level at most.	would like to leave the title of the publication unchanged. As an example, we would like to cite the article of Professor Elhenawy (doi:10.1128/mBio.00909-14). She studies only a few types of Bacteroides, but introduce a generic term into the title of the publication.	
Many sources cited in the manuscript's introduction incorrectly reference the findings of the original paper. Furthermore, some claims lack supporting citations. Here are some examples: Page 2 lines 27-28: IBD therapy does not primarily aim to reduce inflammation with hormones and antibiotics. Perhaps here they are referring to corticosteroids, but this is generally for short-term use and not a mainstay of IBD therapy.	“Standard IBD therapy aims to reduce inflammation with hormones and antibiotics.” We modified to “Standard IBD therapy aims to reduce the acute phase of inflammation with hormones and antibiotics”. We have made a clarification, since hormonal therapy is basic at the stage of the acute inflammatory process. Then, as a rule, non-hormonal therapy is used. In our research, we focus on the possibility of using probiotics with anti-inflammatory properties to prevent the occurrence of the acute stage of IBD. We also change citation: 6.Wright EK, Ding NS, Niewiadomski O. Management of inflammatory bowel disease. Med J Aust. 2018;209(7):318-323. https://doi:10.5694/mja17.01001 7.Seyedian SS, Nokhostin F, Malamir MD. A review of the diagnosis, prevention, and treatment methods of inflammatory bowel disease. J Med Life. 2019;12(2):113-122. https://doi:10.25122/jml-2018-0075 - абзац про History of Disease Treatment	Line 58-59 Line 485-489
Page 3 lines 1-2: Although the cited source Pobeguts et. al 2020 is relevant for the virulence properties (*Escherichia coli* colonizing macrophage) developed in CD isolate *E. coli* under propionate-rich media, no where in the source provide general evidence for Bacteria developing pathogenic phenotype to inflammatory intestinal environment.	Pobeguts's research, including the one we refer to in this statement, are devoted to pathogenic E-Coli. Author provide convincing evidence for the change of the E-Coli phenotype to pathogenic under the influence of altered cultivation conditions. The author studies the conditions for the formation of Crohn's disease in the presence of pathogenic E-Coli, as well as the change of its phenotype from adhesive to adhesive-invasive during the formation of inflammation in the intestine. According to this study, we suggest that bacteria can exhibit new properties, including	Line 63-64

	virulent ones under inflammatory conditions. However, we have rephrased our statement: Bacteria may exhibit new virulent properties when they adapt to chronic inflammation in the intestine	
Page 3, lines 10-11: I remain unconvinced by the authors' reasoning that *Bacteroides* species cannot be used as a single component of probiotics, even after reviewing the original source, Sun et al., 2019.	You are right, this statement is not correct. We have rephrased our statement: The Bacteroides fragilis may be used as a mono probiotic therapy or as part of a complex probiotic in the future, but it is important to continue the research of its biological properties.	Line 72-74
Page 3, lines 18-20: the authors state, "multi-component OMVs are expected to be more effective than isolated PSA in reducing inflammation and subsequent microbial colonization." However, the cited source, Mazmanian et al., 2008, provides no evidence supporting this; the source only discusses the effect of *B. fragilis* isolated PSA on colonic inflammation, with no mention of the complementary effect of OMVs. Mazmanian papers on OMVs were published in 2012 and later.	We have replaced the source of the citation. Professor Elhenawy does not directly state that vesicle enzymes can determine the greater effectiveness, compared to the individual vesicle components. But we assume that the possibility of enhancing the therapeutic effect of whole vesicles, in relation to individual vesicle components, exists. 17. Elhenawy W, Debelyy MO, Feldman MF. Preferential packing of acidic glycosidases and proteases into Bacteroides outer membrane vesicles. mBio. 2014;5(2):e00909-e914. Published 2014 Mar 11. https://doi:10.1128/mBio.00909-14	Line 526-528
Page 3 lines 24-27: there's a lack of citation.	Necessary citations were added 22.Martin JC, Bériou G, Josien R. Dextran Sulfate Sodium (DSS)-Induced Acute Colitis in the Rat. Methods Mol Biol. 2016;1371:197-203. https://doi:10.1007/978-1-4939-3139-2_12 23.Gaudio E, Taddei G, Vetusch A, et al. Dextran sulfate sodium (DSS) colitis in rats: clinical, structural, and ultrastructural aspects. Dig Dis Sci. 1999;44(7):1458-1475. doi:10.1023/a:102662032285	Line 542-545
Page 3 lines 27-32: the manuscript states that "... reproduced a severe DSS response when introduced to germ-free mice, thereby confirming the causal relationship between these species and the severity of DSS colitis." However, the cited	Here we provide a direct quote from the author. Please follow this link to read the author's own story about the study he conducted. https://www.nature.com/articles/s41684-022-00975-4	

source, Forster et al. 2022, only suggests a "potential" cause for DSS endpoint variability. The observed difference in germ-free mice monocolonized endpoints is insufficient to validate a causal conclusion. Also, it is important to mention that the source's identified species are mouse specific.		
Page 3 lines 32-33: there is a lack of citation for why significant changes in microbiota composition are not expected from a DSS model.	We decided to remove our conclusion, since we cannot provide reliable sources and rather imply that the DSS itself should not have a significant effect on the composition of the microbiota. While the indirect effect during the colitis formation should be expressed significantly. Actually, further in the text we conclude that in the DSS-induced colitis model, a significant change in the composition of the microbiota is observed.	
Page 4 lines 20-24 (OMV isolation): The authors' initial filtration step uses a 0.45 μm membrane, which is adequate for removing cells but may not efficiently remove all cell debris, which could contaminate the OMV preparation. As seen in Supplementary Figure S1, the OMV yield appears low, and potential contaminants in the OMV preparation could significantly undermine the paper's claim. A secondary filtration step could be considered prior to ultracentrifugation using a smaller pore size filter, such as 0.22 μm, to ensure more rigorous clearance of finer particulate matter. Also, vesicle pellet was resuspended in distilled water or 150 mM NaCl, however, using isotonic solutions that mimic physiological conditions (e.g., phosphate-buffered saline) could help maintain vesicle integrity better than distilled water.	Dear colleague, before using vesicles for colitis therapy, we conducted many experiments with different isolation and purification protocols. In particular, we conducted proteomic and microbiological analysis of vesicle preparations obtained using a 0.22 μm filter before and after the ultracentrifugation stage. We did not find any significant difference and chose the method that we considered the most convenient. Moreover, our previous researches were carried out according to this protocol. As for the quality of the vesicle preparation in the micrograph presented in the publication, we only wanted to reflect the two-membrane structure of the vesicles, significantly diluting the preparation. In our previous publications, we presented a significant number of electron micrographs, which clearly show that a significant number of vesicles are isolated. In addition, in one of our publications, we provide accurate quantitative metrics, including the concentration of vesicles in the preparation and the determination of particle sizes by the NTA method. doi: 10.3389/fmicb.2023.1164877. doi: 10.1038/s41598-017-05264-6. doi: 10.3389/fcimb.2017.00002. doi: 10.3390/ijms25063295.	

Page 4 lines 25 (OMV isolation): The method mentions quantifying OMVs by dry precipitation but does not provide details on how this is carried out or why it was chosen. Dry weight can be a useful measure of total biomass but may not accurately reflect vesicle concentration or integrity. More commonly, protein content or nanoparticle tracking analysis might offer more detailed insights into vesicle quantity and size distribution.	We reviewed a significant number of publications before choosing the dry matter dose assessment method. We had all the metrics mentioned in the literature, including protein concentration or particle concentration. However, we found the dry matter method to be more reliable when it comes to calculating the dose for a large number of animals over a long observation period. How it was done: after the ultracentrifugation stage, the sediment was weighed on a high-precision analytical balance and then dissolved in an isotonic solution. The weighing process took place in a very short period of time, and we assessed the integrity of the vesicles using electron microscopy. Below you can see the publications according to which we chose the dry residue dose assessment method. Note that the authors do not indicate how exactly they quantify vesicles for animal feeding. doi.org/10.3389/fimmu.2021.777147 doi:10.3390/nu13103319 doi:10.1128/spectrum.01368-22	
Page 5 lines 13-14 (Animals): The methods section mentions an alternating schedule between DSS and water during the 20-day maintenance phase for the DSS and DSS+OMV groups. However, it lacks precise details about the DSS administration schedule, such as frequency and duration of DSS exposure. This lack of specificity could lead to variability in the degree of induced inflammation between subjects, thereby affecting the consistency of the experimental conditions.	We conducted the experiment according to previously published schemes of acute and chronic inflammation formation with minor modifications. Days 1-5, DSS and DSS+OMV groups receive 3% DSS; Days 6-10, water for all groups; Days 11-16, the DSS group receives DSS solution, the DSS+OMV group receives DSS solution and vesicles; Days 17-20, the DSS group receives water, DSS+OMV receives water and vesicles. The model is very dynamic and flexible, depends on the animal's susceptibility, the quality of the reagent, and allows changing the conditions depending on the severity of colitis development and the goals of the experiment. doi:10.3748/wjg.v23.i33.6016 doi:10.1038/nprot.2017.044	Line 137-140

Page 5, lines 15-17 (Animals): The manuscript fails to provide information about the frequency of OMV administration (e.g., daily, every other day). Additionally, it does not specify the route of OMV administration, which is crucial because the route (oral, intravenous, etc.) can significantly influence the therapeutic efficacy of OMVs. It would be essential to ensure that the volume and method of administration match exactly those of the treatment groups to maintain consistent handling stress across all groups.	Mice were gavaged orally with the DSS solution. We mentioned it on Material and methods section.	Line 135
Page 9, lines 8-9, and Figure 1A K3: the authors claim that "several animals from the control group were characterized by non-extended areas of goblet cell loss". However, it's unclear how these 3 histology slides were selected from the 10 samples per group. Additionally, there are no figures or data provided for readers to assess the "several" cases, with only 1 case being presented per group (K3, DSS3, DSSOMV-3).	We have thought through this issue in detail. The fact is that counting goblet cells in the mucous membrane of control animals is not difficult. However, difficulties arise with quantitative counting of goblet cells in experimental groups. Counting cells near the site of destruction of the mucous membrane is not indicative, since when normalizing for the length of the mucous membrane, swelling of the mucous membrane and a decrease in high crypts are not taken into account. Thus, we came to the conclusion that quantitative presentation of data is impossible due to the lack of standardization of counts between animals of different groups. In the text we additionally indicated that in the control group, after staining, goblet cells are well visualized, however, in the experimental group, due to significant destruction of the mucous membrane, we cannot accurately count the number of goblet cells.	Line 272-274
Page 10, lines 9-10: The statement regarding the re-introduction of DSS is unclear because the methodology does not explicitly mention this.	We have detailed the DSS-induced colitis protocol in material and methods section to understand the stages of DSS administration Days 1-5, DSS and DSS+OMV groups receive 3% DSS; Days 6-10, water for all groups; Days 11-16, the DSS group receives DSS solution, the DSS+OMV group receives DSS solution and vesicles; Days 17-20, the DSS group receives water, DSS+OMV receives water and vesicles.	Line 137-140
Page 10, lines 16-17: (referencing Table 1 and Figure 2), the	For statistical analysis, metabolites that occur in no less than 60% of all	

text mentions, "at least 60% of the detected stable compounds were used for the metabolomic profiles comparison." However, the significance of the 60% is unclear, and the inclusion/exclusion criteria for Table 1 and Figure 3 further compound the confusion. Are all figures from Figure 2 onwards solely based on the metabolite sample size of n=263 as mentioned in Table 1?	samples were selected to filter potential inaccuracies in compound identification after HS-GC/MS analysis. The 60% threshold was chosen based on the percentage content of all detected substances. Table 1 contains all metabolites that meet this criterion and are included in further statistical analysis. Table in Figure 3B (prior to adjustments) lists compounds that show significant differences after Mann-Whitney and FDR tests.	
Why does the PCA and OPLS DA data display only 8 samples per group, in contrast to the initial sample size per group of 10 Furthermore, why does the heatmap scale range from -1 to -12? It appears the heatmap is missing a label, which I assume indicates log-fold changes. If that's the case, the authors should consider implementing a better normalization strategy, improving the color scale, or providing a more detailed description in either explaining the results or in the methods section. The blank sections of the heatmap create confusion; it's unclear if these values are NAs or if they represent a log fold value of approximately -6.5. The methodology for the correlation analysis is unclear, and was not clear in delivery how to interpret the meaning of the paper's findings.	To construct the OPLS-DA plot, the PLS Regression model from the scikit-learn library was used, so 20% of the data allocated for model training. The PCA plot was constructed using a sample size of 10. The color scale on the heatmap corresponds to the natural logarithm of compound concentrations. Empty sections represent NA values. Thus, the heatmap displays differences between mean compound concentrations across different groups. Correlations between microbiota and metabolite levels were analyzed by calculating the Spearman correlation coefficient as described in Methods. No grouping of samples into comparison groups was used in this calculation. We have also reworded the text to clarify the findings made by correlation analysis: Numerous significant correlations were found between microbiota composition and metabolomic compounds (Figure 7). The alpha diversity indices and the relative abundance of Saccharibacteria and Acetivibrio were found to be positively correlated with phenol and pentanoic acid levels. In addition, as mentioned above, the alpha diversity indices and the abundance of Saccharibacteria and Acetivibrio were increased in the control group compared to the DSS and DSS+OMV (Figure 5D, 6). The microbiota of DSS group is characterized by an increased representation of Lactococcus and Romboutsia (Figure 6), and it was also found that their increase is associated with decreased phenol and pentanoic acid concentrations	Line 378-388

	(Figure 7). Thus, taking into account the observed associations between the composition of the microbial community and the studied metabolites, a decrease of phenol and pentanoic acid levels may be a marker of dysbiosis associated with the DSS effects.	
Page 11, lines 16-18: Authors mention significant depletion of the metabolic profile in the DSS group compared to the control and DSS OMV groups however, figure 2C is not enough to support this claim. There needs to be a supplementary figure or table or some numbers to support this. Also, the claim around the OMV's tendency to "restore the metabolite profile is also weak.	All graphical data are plotted according to the results obtained during the metabolomic analysis. All obtained values, including metabolites, revealed in the three groups are available in the supplementary materials (Table S2).	
Page 11, lines 23-27: the paper states that "most of the significant differences were found between the control and DSS OMV groups" regarding metabolites. However, Figure 3C appears to contradict this statement looking carefully at that the figure legend (heatmap scale). Furthermore, the statement "It can be assumed that a new equilibrium of the relative amounts of individual metabolites can be observed after OMVs treatment" makes a strong assumption without substantial evidence.	We draw our conclusions regarding the events of day 20, where differences between all groups are clearly visible, both at the metabolome and metagenome levels. The statistical methods used confirm the statistical significance of the differences identified.	
Page 13, lines 12-18:I am unsure how the authors can draw comparative inferences from Figure 7, such as the "predominance of" x bacteria being negatively or positively "correlated with" metabolite y when comparing group A and group B. This concern is significant, as this figure is supposedly one of the main supports for the paper's claim.	We have changed this phrase. Bacterial representation in the comparison groups and correlations of microbiota and metabolites were calculated independently. Numerous significant correlations were found between microbiota composition and metabolomic compounds (Figure 7). The alpha diversity indices and the relative abundance of Saccharibacteria and Acetivibrio were found to be positively correlated with phenol and pentanoic acid levels. In addition, as mentioned above, the alpha diversity indices and the abundance of Saccharibacteria and Acetivibrio were increased in the control group compared to the DSS and DSS+OMV (Figure 5D, 6). The microbiota of DSS group is characterized by an increased representation of Lactococcus and	Line 378-388

	Romboutsia (Figure 6), and it was also found that their increase is associated with decreased phenol and pentanoic acid concentrations (Figure 7). Thus, taking into account the observed associations between the composition of the microbial community and the studied metabolites, a decrease of phenol and pentanoic acid levels may be a marker of dysbiosis associated with the DSS effects.	
Page14, lines 2-4: The claim that the use of OMVs is more beneficial than isolated PSA due to the presence of enzymes improving digestive function lacks strong evidence. We need to specify what "digestive function" means and provide specific proof that the enzymes in OMVs directly contribute to better results in IBD treatment beyond delivering PSA.	Previously, studying the Professor Elhenawy research, we concluded that a significant proportion of hydrolases in Bacteroides vesicles are actively involved in human digestion processes. Therefore, we assume that not only PSA, but also a spectrum of enzymes (which we also observed during our previous proteomic experiments) can have a combined beneficial effect in the treatment of inflammatory bowel diseases. But we modified our statement: We suggest that the use of OVM for the treatment of IBD is more justified than the use of isolated PSA, since the vesicles contain a significant number of enzymes that help improve digestive function. doi:10.1128/mBio.00909-14 doi: 10.1038/s41598-017-05264-6.	
Page14, lines 20-22: The discussion implies that OMVs can aid in restoring the original gut homeostasis, but lacks substantial experimental evidence to establish a clear causal link between OMV treatment and specific changes in microbiota composition and gut mucosal healing. This claim should be supported by more detailed data or mechanistic insights. While the results highlight differences between the experimental groups, they don't provide sufficient evidence to support the "recovery" claim.	A significant amount of obtained histological data indicates high rates of mucosal recovery against the background of OMVs administration. Analysis of the microbiota on the 20th day of the experiment convincingly shows the restoration of the representation of individual bacterial species to control values. We assume that vesicles contribute to a decrease in the inflammatory response, which has already been shown before us, thereby accelerating the rate of mucosal recovery and the composition of the microbiota. We have added more representative histological data to the supplementary materials to confirm our findings and mentioned it in manuscript.	Line 261, 263, 270 Supplementary information file
Page 4 lines 14/16 (Bacterial strain and growth conditions):	The criterion for the stationary phase was a plateau when measuring the	Line 110-111

The protocol mentions cultivation until stationary phase but does not specify how this phase is determined. For experimental reproducibility, it is crucial to define the stationary phase more clearly, perhaps through optical density measurements at a specific wavelength (e.g., OD600) or via viable cell counts.	optical density at 600 nm. The corresponding information has been added to the manuscript.	
Page 5 line 14-19: While the paper includes an ethics statement, the use of chloroform for euthanasia is still concerning. Chloroform can cause respiratory distress and inconsistent times to loss of consciousness, potentially leading to unnecessary suffering and stress. This could also mildly impact experimental outcomes, especially those involving inflammatory processes. Current guidelines and standards recommend using methods like CO2 asphyxiation followed by cervical dislocation.	Thank you very much for your important note. There was a mistake in the description of the euthanasia method. Isoflurane (4%, 2 L/min) was used for this experiment. We have made corrections in the manuscript and once again agreed with our ethics committee on the error made during the writing of the publication.	Line 140-141 Line 145-146
Page 7 15-18 (Metabolome data processing): Assuming that the raw data do not follow a normal distribution without conducting any formal test for normality, such as the Shapiro-Wilk test, could be an issue. Although it's practical to use non-parametric tests like the Mann-Whitney test for non-normal distributions, it's vital to verify the type of distribution through appropriate tests. The methodology discussed the use of both the Mann-Whitney test and ANOVA after a natural log transformation. It would be helpful to provide a clearer rationale for the choice of each test based on the data distribution and to apply corrections for multiple comparisons.	The Shapiro-Wilk test for normality of the data was conducted. The data did not follow a normal distribution. Therefore, the decision was made to use the non-parametric Mann-Whitney test. The Benjamini-Hochberg correction for multiple comparisons was applied in all statistical tests.	
Figure 1B/C: The plots representing histological index and DAI of the distal intestine show extremely wide confidence	In Figure 1B/C, the p-values resulting from statistical comparisons of mean values with Benjamini-Hochberg correction for multiple	

intervals with significant overlap between groups. A statistical comparison of mean values is needed, along with the calculation of p-values.	comparisons are presented. For the histological index, a non-parametric Mann-Whitney test was conducted. For DAI, comparisons were conducted using the Kruskal-Wallis test for each day except days 1 and 3 (where DAI values were zero for the DSS group). Values with '*' denote p-values <0.001.	
Page 11, lines 20-23: It is unclear how the compounds in the table shown in Figure 3B were selected.	The table shows the compounds reliably detected in all samples of all experimental groups	
Page 13, lines 5-11: Figure 6 has low resolution and cannot be interpreted.	The image quality has been improved	Line 655
Page 2 line 5: Could the authors define what they mean by "intermediate" in the phrase "microbiota composition of the DSS OMV group was intermediate between the control and DSS groups"?	With this statement we wanted to reflect the visual component when comparing the metagenomic data of the three groups. In particular, in the control group and the DSS OMV group, there is a significant variation in microorganisms, while in the DSS group there is a significant gap in the representation and number of species. However, when evaluating the data obtained, we see that some species that were not identified in the DSS group and previously appeared in the control group reappear in the DSS OMV group. We concluded that the microbiota tends to recover, but it takes more time to form a primary or control intestinal microbiome.	
Page 2 lines 9/11: I presume "Bacteroide fragilis" is a typo for Bacteroides fragilis.	Necessary correction has been made	Line 41
Page 8 line 23: Versions for Porechop and NanoFilt softwares are missing.	Porechop v0.2.3 NanoFilt V2.8.0	Line 244
Figure 3A: Labels unclear.	The Labels quality has been improved	Line 633
The authors should define "VOCs," which I assume stands for Volatile Organic Compounds, somewhere in the paper	The term was decrypted	Line 630

Re: Spectrum00636-24R2 (Bacteroides vesicles promote functional alterations in gut microbiota composition)

Dear Dr. Natalya B. Zakharzhevskaya:

I would again like to apologize for the back and forth process and delay after receiving Reviewer 2's comments after getting back to you for revisions earlier. Thank you for carefully addressing both Reviewers remarks and suggestions, I feel like it helped improve the manuscript. I hereby would like to congratulate you on the acceptance of your paper for publication in Spectrum!

Your manuscript has been accepted, and I am forwarding it to the ASM production staff for publication. Your paper will first be checked to make sure all elements meet the technical requirements. ASM staff will contact you if anything needs to be revised before copyediting and production can begin. Otherwise, you will be notified when your proofs are ready to be viewed.

Sincerely,
Jan Claesen
Editor
Microbiology Spectrum